# HYPERFLOW: GRADIENT-FREE EMULATION OF FEW-SHOT FINE-TUNING

## ABSTRACT

While test-time fine-tuning is beneficial in few-shot learning, the need for multiple backpropagation steps can be prohibitively expensive in resource-constrained environments or end devices. To address this limitation, we propose a computationally efficient test-time adaptation approach that emulates gradient descent without computing gradients. Specifically, we formulate gradient descent as an Euler discretization of an ordinary differential equation (ODE) and train a lightweight auxiliary network to predict the task-conditional drift using only the few-shot support set. The adaptation then reduces to a simple numerical integration (*e.g.,* via the Euler method), which requires only a few forward passes of the auxiliary network—no gradients or forward passes of the target model are needed. In experiments on cross-domain few-shot classification using the Meta-Dataset and CD-FSL benchmarks, our method significantly improves out-of-domain performance over the non-fine-tuned baseline while incurring only 6% of peak memory and 0.1% of the FLOPs of standard fine-tuning, thus establishing a practical middle ground between direct transfer and fine-tuning approaches.

## 1 INTRODUCTION

Few-shot classification aims to recognize novel classes of images using only a handful of labeled examples per class (Fei-Fei et al., 2006; Lake et al., 2015; Vinyals et al., 2016; Ravi & Larochelle, 2017). Unlike conventional supervised learning approaches that use an extensive amount of labeled images for pre-defined tasks to train specialized models, few-shot learning approaches focus on adapting a pretrained model to novel tasks using a few examples, where the target classes and image domains can vary at test time. This paradigm is vital for real-world scenarios where the image domain is distinctive and the labeled data is scarce or expensive to obtain, such as medical imaging (Kotia et al., 2020), plant disease classification (Argüeso et al., 2020), and remote sensing (Sun et al., 2021). Even in the era of increasing general-purpose foundation models (Bommasani et al., 2021), few-shot learning is still beneficial to achieving optimal performance in domain-specific contexts (Liu et al., 2024; Xu et al., 2024; Madan et al., 2025).

Recent advances in few-shot classification often rely on an additional *fine-tuning* stage using the few examples given at test time. Thanks to the simplicity and effectiveness of their design, embedding-based algorithms (Vinyals et al., 2016; Snell et al., 2017; Bateni et al., 2020) have been predominant in few-shot classification tasks. These methods classify images by learning a shared embedding space and assigning labels based on distances or similarities to reference points (*e.g.,* class prototypes or exemplars) within that space. While these approaches were originally designed to directly transfer a pretrained feature extractor to novel classes without adaptation, recent studies have consistently shown that additional fine-tuning at test time significantly improves performance, particularly when substantial domain discrepancies exist between training and testing scenarios (Guo et al., 2020; Phoo & Hariharan, 2021; Islam et al., 2021; Oh et al., 2022; Hu et al., 2022b).

However, despite its effectiveness, fine-tuning incurs high computational costs, both in memory and time, making it impractical for many real-world few-shot learning scenarios such as resource-constrained environments or end devices. This high computational overhead primarily results from the repeated backpropagation required during gradient descent. Therefore, achieving a balance between direct transfer and fine-tuning necessitates a computationally efficient adaptation mechanism that avoids gradient-based optimization entirely. This perspective shares a high-level inspiration with meta-learning methods (Hu et al., 2020; Zhang et al., 2022; Du et al., 2023; Zhang et al., 2024)

that propose to replace the inner loop with a gradient-free process, such as synthetic gradients (Jaderberg et al., 2017), Neural ODE (Chen et al., 2018), and diffusion (Ho et al., 2020). However, these methods primarily focus on adapting only the linear classifiers or *prototypes*, and they still require backpropagation of the target model in the outer loop. On the other line of work, parameter generation methods (Knyazev et al., 2021; Peebles et al., 2022; Schürholt et al., 2022; 2024) propose to generate more general parameters. However, most existing methods primarily focus on modeling parameter distributions for a pre-defined task with different architectures or training configurations, and relatively little attention has been paid to few-shot learning scenarios.

We propose *HyperFlow*, a model-agnostic test-time adaptation mechanism for adapting general *parameters* of the target model without computing any gradients. To achieve this, we introduce an auxiliary, lightweight conditional drift network that learns the parameter dynamics of the ordinary differential equations (ODEs) induced by the gradient descent (*i.e.,* gradient flows). As the gradient flow depends on the tasks, we amortize them by conditioning the drift network on a few support examples. Since directly learning the ODEs in the full parameter space of modern neural networks is infeasible, we adopt a parameter-efficient fine-tuning (PEFT) approach (Zaken et al., 2022), which reduces the total parameters that need to be updated and thereby enables scalable training and inference. In addition, we approximate the continuous gradient flows by interpolating the discrete gradient descent trajectories collected from an offline simulation on the meta-training dataset, enabling efficient training of the conditional drift network. As a result, the fine-tuning stage can be replaced with a lightweight ODE-solving process (*e.g.,* Euler method) that involves only a few forward passes of the conditional drift network. Importantly, both the training and inference procedures of HyperFlow are decoupled from the target model, thus scaling well to large target models and incurring negligible computational overhead compared to fine-tuning approaches.

We demonstrate our approach in cross-domain few-shot classification settings using widely adopted benchmarks such as Meta-Dataset (Triantafillou et al., 2020) and CD-FSL (Guo et al., 2020). When applied to the state-of-the-art few-shot classification approach (Hu et al., 2022b), HyperFlow can significantly improve the out-of-domain generalization by only using $6\%$ of peak memory and $0.1\%$ of FLOPs compared to conventional fine-tuning. As a result, HyperFlow offers a range of trade-offs between the computation cost and the performance and thus suggests the practical middle-ground between the direct transfer and fine-tuning approaches.

## 2 PRELIMINARY

**Problem Setup**  Few-shot classification (FSC) aims to recognize novel classes of images using only a few labeled examples per class. To achieve this, a model is first trained on a base dataset $\mathcal{D}_{\text{base}} = \{(x_i, y_i)\}_{i=1}^{N_{\text{base}}}$, often called as "meta-training" dataset, which consists of many labeled examples for known base classes $\mathcal{C}_{\text{base}}$. During testing, the model is evaluated on multiple classification tasks involving novel classes $\mathcal{C}_{\text{novel}}$, where $\mathcal{C}_{\text{base}} \cap \mathcal{C}_{\text{novel}} = \phi$. To learn the novel classes of each test task $\mathcal{T}$, the model receives a small set of labeled examples $\mathcal{S}_{\mathcal{T}} = \{(x_j, y_j)\}_{j=1}^{N_{\mathcal{T}}}$ called the *support set*. Then, the model is tested to classify unlabeled images within the same task, called the *query set*. In a conventional $N$-way $K$-shot setting (Vinyals et al., 2016; Finn et al., 2017), each task consists of $N$ classes, and $K$ labeled images per class are given as the support set.

In this paper, we consider a more challenging *cross-domain* few-shot classification problem, where the testing images originate from domains different from those of the base dataset $\mathcal{D}_{\text{base}}$ (*e.g.,* domain shifts from natural images to X-ray). In this setup, the model must equip a strong generalization ability and a flexible adaptation mechanism to address both novel classes and out-of-domain images.

**P>M>F Pipeline for FSC**  While various approaches have been proposed to tackle the FSC problem, a common training pipeline involves one or more of three distinct stages—Pre-training, Meta-training, and Fine-tuning—collectively known as P>M>F (Hu et al., 2022b). In the pre-training stage, general-purpose representations are learned using backbones trained with self-supervised methods on large-scale datasets (*e.g.,* the DINO backbone (Caron et al., 2021) pre-trained on ImageNet (Deng et al., 2009)). During the meta-training stage, the model acquires prior knowledge about the underlying task distribution, typically by simulating few-shot learning episodes using the base dataset $\mathcal{D}_{\text{base}}$ (Vinyals et al., 2016). Finally, the fine-tuning stage further adapts the model to test tasks using the provided support set, enabling flexible adaptation to novel classes and unseen domains. During the fine-tuning stage, parameter-efficient fine-tuning (PEFT) techniques (Zaken et al., 2022; Hu et al., 2022a) can be applied to reduce over-fitting to the support set.

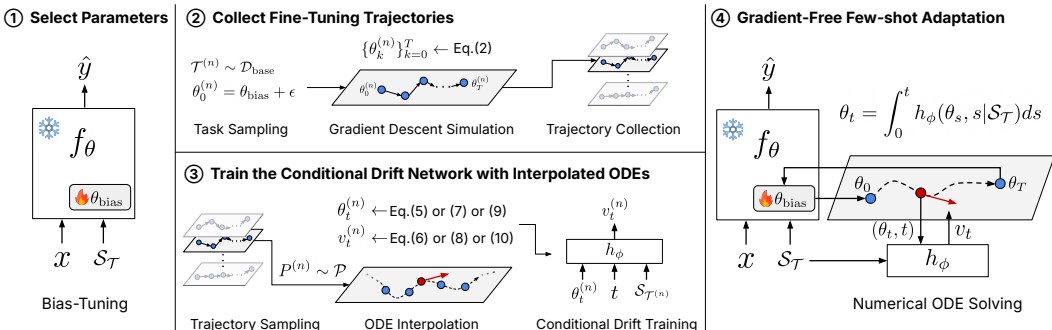

Figure 1: An overview of HyperFlow. (1) For scalable computation in the parameter space, we select the bias parameters of the target model $f_\theta$ to be updated. (2) We collect $T$-step fine-tuning trajectories by simulating gradient descent on the bias parameters, using the base dataset where the target model has been trained. (3) After collecting the trajectories, we continuously interpolate them to train the conditional drift network $h_\phi$ on the continuous time interval $[0, T]$. (4) At test time, we employ a numerical ODE solver (*e.g.,* Euler method) with a few forward passes of $h_\phi$ to adapt the bias parameters on the support set.

## 3 METHOD

In this work, we focus on the *fine-tuning* stage of the training pipeline of few-shot classification (FSC). While it has been consistently reported that fine-tuning is beneficial for FSC (Guo et al., 2020; Phoo & Hariharan, 2021; Islam et al., 2021; Oh et al., 2022; Hu et al., 2022b), especially in cross-domain scenarios, it necessitates the computation of expensive backpropagation throughout the network. This can be problematic in practical applications of FSC where only low resources are allowed for adaptation. For an efficient few-shot adaptation, we propose HyperFlow, which replaces the fine-tuning stage with a much cheaper process. See figure 1 for an overview of our method.

### 3.1 HYPERFLOW: EMULATING GRADIENT FLOWS

Fine-tuning involves gradient descent on the model parameters. Let $f_\theta$ be the model with parameters $\theta$ and $\mathcal{L}$ be a loss function (*e.g.,* cross-entropy). Given a test task $\mathcal{T}$ with a corresponding support set $\mathcal{S}_\mathcal{T} = \{(x_i, y_i)\}_{i=1}^{N_\mathcal{T}}$, the objective of few-shot adaptation is to minimize the loss over the support:

$$L_\mathcal{T}(\theta) = \sum_{(x_i, y_i) \in \mathcal{S}_\mathcal{T}} \mathcal{L}\left(f_\theta(x_i), y_i\right). \tag{1}$$

This is achieved by iterative gradient descent updates on $\theta$ from its initialization $\theta_0 = \theta_{\text{init}}$ as follows:

$$\theta_k \leftarrow \theta_{k-1} - \lambda \cdot g\left(\nabla_\theta L_\mathcal{T}(\theta_{k-1})\right), \quad k = 1, 2, \cdots, T, \tag{2}$$

where $\theta_k$ denotes the parameters at $k$-th iteration, $\lambda$ denotes the learning rate, and $g$ is an optimizer (*e.g.,* Adam (Kingma & Ba, 2015)). However, in resource-constrained environments, computing the gradient $\nabla_\theta L_\mathcal{T}(\theta)$ can be burdensome, especially when the network $f_\theta$ is deep and large.

Our key idea is to emulate the gradient descent updates (Eq. 2) with a *gradient-free* ODE-solving process. As already remarked by prior studies (Merkulov & Oseledets, 2020; Elkabetz & Cohen, 2021; Bu et al., 2021), the gradient descent updates can be seen as an Euler discretization of the corresponding ordinary differential equation (ODE), which is also known as the gradient flow (Santambrogio, 2017). For example, the gradient flow of task $\mathcal{T}$ with the simplest gradient descent optimizer (*i.e.,* $g(\theta) = \theta$) can be expressed as follows:

$$\frac{d\theta(t)}{dt} = -\frac{\partial L_\mathcal{T}(\theta(t))}{\partial \theta}. \tag{3}$$

Inspired by Neural ODEs (Chen et al., 2018), we approximate the ODE function using a drift network $h_\phi$. Then, we can emulate the gradient descent by a numerical integration method (*e.g.,* the Euler method), which only requires several forward steps of $h_\phi$.

Since the gradient flow differs by tasks, we employ a *conditional* drift network $h_\phi(\theta, t|\mathcal{S}_\mathcal{T})$ to predict the task-specific drift $\frac{d\theta(t)}{dt}$ of task $\mathcal{T}$ conditioned on the support set $\mathcal{S}_\mathcal{T}$. To train the conditional

drift network, we perform an extra training stage using the base dataset as follows:

$$\min_{\phi} \ \mathbb{E}_{t,\mathcal{T}} \left[ \left\| h_\phi(\theta_t, t | \mathcal{S}_\mathcal{T}) - \left( -\frac{\partial L_\mathcal{T}(\theta(t))}{\partial \theta} \right) \right\|^2 \right], \tag{4}$$

where the timestep $t$ is uniformly sampled from $[0, T]$ and the training task $\mathcal{T}$ is sampled from the base dataset $\mathcal{D}_{\text{base}}$. As the base dataset is often assumed to be diverse enough to cover the underlying task distribution, the trained drift network $h_\phi$ can generalize to novel tasks. In Section 5.4, we study how the diversity of the base dataset affects the generalization ability of $h_\phi$.

### 3.2 TARGET PARAMETERS SELECTION

A straightforward challenge in learning the gradient flow with a drift network is that the number of target network parameters is often too large. For example, vision transformers (Dosovitskiy et al., 2021), a widely used neural network architecture in computer vision, typically have millions to billions of parameters, making it nearly infeasible to directly treat them as inputs and outputs of another neural network. However, recent studies on parameter-efficient fine-tuning (PEFT) show that updating a tiny set of parameters is sufficient to modulate large pre-trained backbones (Rebuffi et al., 2017; Zaken et al., 2022; Hu et al., 2022a). Therefore, we reduce the number of parameters considered in the gradient flows by selecting a tiny subset of the whole parameters $\theta$.

Specifically, we adopt bias-tuning (Zaken et al., 2022) that updates only the bias parameters of the model $f_\theta$. In the few-shot learning context, bias-tuning has been shown to be extremely parameter-efficient and robust to over-fitting for adapting large transformers to unseen vision tasks (Kim et al., 2023; 2024; Chen et al., 2023). Also, it adds no extra computation overhead to the original model during inference. By further selecting a subset of bias parameters (*e.g.,* those from qkv-projection layers of the attention blocks), we can reduce the number of target parameters to be sufficiently tangible for the drift network (*e.g.,* thousands), without losing much adaptability at the few-shot adaptation phase as we show in Section 5.2.

Once we have selected the feasible number of target parameters, another benefit of emulating gradient descent with the conditional drift network is that the computation involved in few-shot adaptation no longer depends on the architecture of the original network $f_\theta$. For example, in conventional fine-tuning, even if we update a small subset of parameters, such as bias parameters, we still need to backpropagate the network until the layers have the parameters. However, the drift network $h_\phi$ can be designed to produce an arbitrary set of parameters at once with a single forward operation. This makes our approach benefit from PEFT techniques in reducing the computation cost, *i.e.,* the parameter-efficiency directly translates to the computation-efficiency. In Section 5.3, we provide a detailed analysis of the computation efficiency of HyperFlow compared to conventional fine-tuning.

### 3.3 DRIFT TRAINING WITH CONTINUOUS ODES

Training the conditional drift network using the continuous-time ODE objective (Eq. 4) requires expensive computation of the gradient $\nabla_\theta L_\mathcal{T}(\theta)$ at arbitrary timestep $t \in [0, T]$ as well as the simulation of the gradient flow until $t$. To make the training process more efficient, we first collect a fixed number of fine-tuning trajectories by simulating the gradient descent on the base dataset. Specifically, for each task $\mathcal{T}^{(n)}$ constructed from the base dataset $\mathcal{D}_{\text{base}}$, a discretized trajectory $P^{(n)} = \{\theta_0^{(n)}, \cdots, \theta_T^{(n)}\}$ is generated by applying $T$ gradient descent steps (Eq. 2) on the initial parameters $\theta_{\text{init}}^{(n)}$. We repeat this process multiple times by randomly perturbing the initialization $\theta_{\text{init}}^{(n)}$, yielding a dataset of trajectories $\mathcal{P} = \{P^{(n)}\}_{n=1}^{N_{\text{traj}}}$. During training, we approximate the continuous gradient flows by smoothly interpolating the discretized points to compute the intermediate parameter $\theta_t$ and its drift $v_t$ at an arbitrary timestep $t$. We propose both linear and nonlinear objective ODEs that interpolate the discretized gradient flows. Below, we omit the superscript of $\theta^{(n)}$ for simplicity.

**Linear Flow (HyperFlow-L)** A linear flow (**HyperFlow-L**) is generated by linearly interpolating the initial and final points of each trajectory (Lipman et al., 2023; Liu et al., 2023). Specifically, given the two endpoints $\theta_0$ and $\theta_T$ of a trajectory, the point $\theta_t$ and its drift $v_t$ (time-derivative) at timestep $t \in [0, T]$ is computed as follows:

$$\theta_t = (1 - t/T) \cdot \theta_0 + t/T \cdot \theta_T, \tag{5}$$
$$v_t = (\theta_T - \theta_0)/T. \tag{6}$$

Figure 2: The the conditional drift network architecture of HyperFlow.

Note that this objective does not use any intermediate points $\{\theta_k\}_{k=1}^{T-1}$. As it promotes the straightness of the learned flow, it encourages the drift network to solve the ODE with only a few Euler steps. However, the straight path emulated by Eq. 8 may deviate from the true trajectory that follows the loss surface, so the flexibility of HyperFlow can be limited by the oversimplified ODEs.

**Nonlinear Flows (HyperFlow-PL & HyperFlow-C)** To cope with more complex loss surfaces, we also propose a nonlinear flow objective that exploits the full trajectories. A straightforward way to extend the linear flow with intermediate points is to interpolate each discrete transition $\theta_k \to \theta_{k+1}$ individually, *i.e.,* define piecewise-linear flows (**HyperFlow-PL**). Specifically, we divide the time interval $[0, T]$ into $T$ segments $[0, 1], [1, 2], \cdots, [T-1, T]$, then linearly interpolate $p_t$ and $v_t$ in each segment $[k-1, k]$ as follows:

$$\theta_t = (k-t) \cdot \theta_{k-1} + (t-k+1) \cdot \theta_k, \tag{7}$$

$$v_t = \theta_k - \theta_{k-1}. \tag{8}$$

However, such non-smooth transitions in the velocity field are difficult to model by the conditional drift network and can cause over-fitting to training trajectories. Thus, we create smooth ODEs that pass through the discrete points using cubic Hermite splines (Hermite, 1863; De Boor, 1978). To compute the interpolated point and its drift, the cubic flow (**HyperFlow-C**) defines local cubic curves passing four neighboring points at each timestep. Specifically, $p_t$ and $v_t$ in the segment $[k-1, k]$ can be computed as follows:

$$\theta_t = a_k t^3 + b_k t^2 + c_k t + d_k, \tag{9}$$

$$v_t = 3a_k t^2 + 2b_k t + c_k, \tag{10}$$

where the coefficients $a_k, b_k, c_k, d_k$ are computed by the four points $\theta_{k-2}, \theta_{k-1}, \theta_k, \theta_{k+1}$ using Catmull-Rom spline (Catmull & Rom, 1974). For interpolating the first and the last segment, we set the sentinels as $\theta_{-1} = \theta_0$ and $\theta_{T+1} = \theta_T$. More details are provided in Appendix B.

### 3.4 CONDITIONAL DRIFT NETWORK ARCHITECTURE

The conditional drift network $h_\phi$ consists of a feature extractor, a task projector, and a drift decoder (see Figure 2 for an illustration). Since the target drift $v_t$ of the ODE depends on the task $\mathcal{T}$, we need an effective task encoding mechanism. To this end, we first encode the support set into a task vector $z_\mathcal{T}$ using the feature extractor and the task projector. We first create class prototypes by encoding the support images using a feature extractor and averaging them for each class. For the feature extractor, we use a ResNet-18 (He et al., 2016) pretrained on ImageNet-1k and freeze it during training. Then, a linear task projector transforms the class prototypes to learnable embeddings, which are further averaged to produce a single task vector $z_\mathcal{T}$. Also, the scalar timestep $t$ is encoded by the sinusoidal embedding and linearly projected to a time embedding, which is concatenated to the task vector.

Conditioned on the task vector and time embedding vector, the drift decoder predicts the drift $v_t$ from the vectorized parameters $\theta_t$. The drift decoder is a shallow MLP whose intermediate features are modulated by adaptive layer normalization (AdaLN) modules (Peebles & Xie, 2023). Each AdaLN module gets the concatenated task vector and time embedding as input, and produces the scale, shift, and gating parameters for each block of the drift decoder. We leave the detailed explanation about the conditioning mechanism of AdaLN and the architectural configurations in Appendix C.

## 4 RELATED WORK

**Metric-Based Few-Shot Learning** Few-shot learning aims to adapt models to handle new tasks with minimal labeled data. A predominant strategy of *few-shot classification*, where each task is defined by a set of image classes (Fei-Fei et al., 2006; Lake et al., 2015), is the metric-based learning that assigns the classes based on the distance on an embedding space learned by a feature extractor (Vinyals et al., 2016; Snell et al., 2017). While early methods of this type typically transfer the

feature extractor to novel classes without additional adaptation, several studies (Guo et al., 2020; Phoo & Hariharan, 2021; Islam et al., 2021; Oh et al., 2022; Hu et al., 2022b) have shown that test-time fine-tuning offers substantial improvements in cross-domain scenarios. In particular, Hu et al. (2022b) introduced P>M>F, a three-stage pipeline—self-supervised pretraining, episodic meta-learning, and test-time fine-tuning—that significantly boosts performance on out-of-domain tasks.

**Meta-Optimizers and Gradient-Based Meta-Learning.** Another branch of few-shot learning research focuses on training meta-optimizers or finding good parameter initializations to quickly adapt to new tasks (Andrychowicz et al., 2016; Finn et al., 2017; Ravi & Larochelle, 2017). Although these methods eliminate hand-crafted optimizers and encourage rapid adaptation, they require gradients for adaptation, increasing computational overhead and hyperparameter sensitivity. Recently, several gradient-based meta-learning methods (Hu et al., 2020; Zhang et al., 2022; Du et al, 2023; Zhang et al., 2024) proposed to replace the inner loop of the bi-level optimization with a gradient-free process, such as synthetic gradients (Jaderberg et al., 2017), Neural ODE (Chen et al., 2018), and diffusion (Ho et al., 2020). However, they still require backpropagation through the target model during the outer loop of training, which restricts their application to adapting only *prototypes* (or linear classifiers) with relatively small target models (*e.g.,* ResNet-12). HyperFlow can be seen as a scalable extension of this line of work for adapting arbitrary *parameters* that does not require backpropagation of the target model in both training and inference.

**Hypernetworks and Parameter Generation** Parameter generation methods aim to generate model parameters using another neural network, often called a *hypernetwork* (Ha et al., 2017). These methods primarily aim to generate neural network parameters that generalize across various architectures or training configurations rather than adapting to novel tasks with a few examples (Peebles et al., 2022; Schürholt et al., 2024; Knyazev et al., 2021; Schürholt et al., 2022). Recently, Jin et al. (2024) and Liang et al. (2024) have proposed few-shot parameter generation methods for natural language processing (and style transfer) and policy learning, respectively. Both rely on latent diffusion models (Rombach et al., 2022) operating in a latent space learned via autoencoders, which often require high computation cost and do not directly focus on efficient adaptation. Our approach, by contrast, explicitly learns the gradient flows in the parameter space, which allows computationally efficient adaptation via a few forward operations of a lightweight conditional drift network.

## 5 EXPERIMENTS

The main advantage of HyperFlow over existing few-shot learning approaches is that it strikes a middle ground between direct-transfer (without fine-tuning) approaches and fine-tuning approaches, offering both improved performance and reduced computational cost to each. As a demonstration, we apply HyperFlow on the P>M>F (Hu et al., 2022b) pipeline, which involves pretraining, meta-training, and fine-tuning stages for cross-domain few-shot classification. Specifically, we replace the fine-tuning stage with HyperFlow and analyze its performance and computational cost.

### 5.1 EXPERIMENTAL SETUP

**Datasets** Following Hu et al. (2022b), we use the Meta-Dataset benchmark (Triantafillou et al., 2020), which consists of ten datasets spanning diverse domains: ImageNet-1k (Inet), Omniglot (Omglot), FGVC-Aircraft (Acraft), CUB-200-2011 (CUB), Describable Textures (DTD), Quick-Draw (QDraw), FGVCx Fungi (Fungi), VGG Flower (Flower), Traffic Signs (Sign), and MSCOCO (COCO). We adopt the 8 in-domain setup, treating the training splits of the first eight datasets as the *base* dataset and evaluating on the test splits of all ten, where Sign and COCO serve as out-of-domain tasks. Test episodes of the Meta-Dataset tasks are constructed in *various-way various-shot*, where both the number of classes and the number of support images per class are randomly chosen. In addition, we include four out-of-domain datasets from the CD-FSL benchmark (Guo et al., 2020): CropDisease, EuroSAT, ISIC, and ChestX. Here, tasks are constructed in 5-way with 5-, 20-, and 50-shots. In all datasets, we evaluate the models with 600 randomly sampled episodes.

**Baselines** We choose the ProtoNet (Snell et al., 2017) variant proposed by Hu et al. (2022b) as our target model. The model is first pretrained on ImageNet-1K with a self-supervised objective. In this study, we use a DINO (Caron et al., 2021)-pretrained ViT-S backbone and a BEiTv2 (Peng et al., 2022)-pretrained ViT-L backbone. In Appendix A, we also provide results with ConvNeXt-S (Liu et al., 2022) backbone to show the backbone generality of HyperFlow. Each backbone is then meta-trained on the eight base datasets with episodic training protocol. Using the meta-trained ProtoNet as a base model, we compare three types of approaches throughout the experiments:

Table 1: Few-shot classification accuracy (%) on Meta-Dataset benchmark. As a gradient-free adaptation mechanism, HyperFlow strikes the middle ground between the direct transfer and fine-tuning methods, while outperforming existing parameter generation methods.

| Model Variant | Average | | Out-of-Domain | | In-Domain | | | | | | | |
|---|---|---|---|---|---|---|---|---|---|---|---|---|
| | OOD | ID | Sign | COCO | Acraft | CUB | DTD | Fungi | Flower | Inet | Omglot | QDraw |
| ViT-S backbone | | | | | | | | | | | | |
| Direct Transfer | 55.24 | 83.31 | 55.35 | 55.12 | 87.26 | 91.40 | 81.63 | 74.16 | 91.06 | 70.78 | 90.18 | 79.98 |
| HyperFlow-L (**Ours**) | 59.12 | 83.42 | 61.78 | 56.46 | 86.76 | 91.18 | 82.30 | 73.83 | 92.27 | 70.69 | 90.15 | 80.14 |
| HyperFlow-C (**Ours**) | 63.11 | 83.33 | 69.49 | 56.72 | 86.74 | 91.07 | 82.05 | 73.82 | 91.50 | 71.48 | 89.89 | 80.12 |
| Head-Tuning | 55.92 | 83.74 | 56.49 | 55.34 | 87.21 | 91.30 | 83.31 | 73.80 | 92.34 | 71.38 | 89.96 | 80.60 |
| Bias-Tuning | 74.06 | 84.69 | 88.35 | 59.77 | 87.49 | 91.21 | 86.43 | 73.88 | 95.68 | 72.10 | 90.62 | 80.14 |
| LoRA-Tuning | 74.97 | 85.17 | 90.58 | 59.36 | 89.61 | 91.81 | 87.12 | 74.00 | 94.81 | 72.08 | 91.83 | 80.09 |
| Full-Tuning | 75.38 | 84.91 | 91.65 | 59.10 | 89.32 | 91.08 | 87.60 | 73.80 | 94.82 | 70.87 | 91.93 | 79.83 |
| ViT-L backbone | | | | | | | | | | | | |
| Direct Transfer | 62.82 | 91.17 | 55.75 | 69.88 | 95.73 | 97.85 | 82.58 | 86.21 | 99.56 | 90.18 | 93.98 | 83.28 |
| HyperFlow-L (**Ours**) | 71.68 | 91.48 | 71.63 | 71.72 | 96.17 | 97.78 | 84.08 | 86.19 | 99.58 | 90.70 | 93.95 | 83.39 |
| HyperFlow-C (**Ours**) | 71.43 | 91.63 | 70.43 | 72.43 | 95.98 | 97.97 | 85.41 | 86.31 | 99.60 | 90.38 | 93.88 | 83.50 |
| Head-Tuning | 65.16 | 92.10 | 60.22 | 70.11 | 96.73 | 97.88 | 87.77 | 86.23 | 99.76 | 90.23 | 93.98 | 84.25 |
| Bias-Tuning | 81.02 | 92.56 | 89.50 | 72.54 | 96.78 | 97.83 | 90.75 | 86.25 | 99.65 | 90.98 | 94.07 | 84.13 |
| LoRA-Tuning | 82.06 | 92.55 | 91.77 | 72.34 | 97.28 | 97.88 | 91.26 | 86.26 | 99.61 | 90.22 | 94.45 | 83.41 |
| Full-Tuning | 81.71 | 92.61 | 91.80 | 71.63 | 97.08 | 97.84 | 90.43 | 86.21 | 99.72 | 90.79 | 94.51 | 84.33 |

- **Direct Transfer** is immediately evaluated without test-time adaptation.
- **Head-Tuning**, **Bias-Tuning**, **LoRA-Tuning**, and **Full-Tuning** fine-tune the parameters of the classifier head (initialized to prototypes), bias, 16-rank LoRA Hu et al. (2022a), and the whole network, respectively. The fine-tuning baselines involve 50 gradient descent steps with the Adam optimizer (Kingma & Ba, 2015), and the learning rate is determined by a few validation samples of each task following Hu et al. (2022b).
- **HyperFlow-L** and **HyperFlow-C** are two representative variants of HyperFlow, replacing the fine-tuning stage of P>M>F pipeline with linear and cubic flows, respectively. For the ODE solving process, we apply the Euler method with at most 50 steps, where the number of steps and the step size are determined similarly to the learning rate selection of the fine-tuning baselines.

**Implementation Details** For HyperFlow and the Bias-Tuning variant, we update the bias parameters of qkv-projection from each attention layer of the backbone transformer, which totals 13,824 and 49,152 parameters for ViT-S and ViT-L, respectively. We train the conditional drift network by collecting fine-tuning trajectories from the base ProtoNet model on the eight training domains of the Meta-Dataset. Specifically, we sample 500 episodes per domain and perform 50 gradient descent steps with the Adam optimizer using a fixed learning rate. For each episode, we apply ten different Gaussian perturbations to the initial parameters, yielding a total of 40,000 trajectories. When training HyperFlow-L, we use only the first and final parameters of each trajectory for interpolation. Further details for offline trajectory collection are provided in Appendix D.

## 5.2 MAIN RESULT

Table 1 and Table 2 present our main results on the Meta-Dataset and CD-FSL benchmarks, respectively. On average, the performance of HyperFlow stands between the Direct Transfer and the Full-Tuning baselines, as expected. Remarkably, we find that incorporating HyperFlow-C into the base model substantially boosts out-of-domain (OOD) performance (Sign, COCO, and all domains in CD-FSL), especially closing the gap between Bias-Tuning in CD-FSL 5-shot tasks. This improvement indicates that HyperFlow is not merely memorizing the fine-tuning trajectories but is instead learning a shared geometry of the underlying loss surfaces across tasks. For in-domain tasks, the base model has already gained sufficient knowledge during the meta-training stage. Thus, applying test-time adaptation (even Full-Tuning) shows limited additional gains over Direct Transfer, which is also consistent with the findings in Hu et al. (2022b).

It is also noticeable that both variants of HyperFlow outperform the simple Head-Tuning in all OOD tasks, while using much lower computing (FLOPs) and comparable peak memory requirement as presented in Figure 3. By comparing two variants of HyperFlow, we observe that HyperFlow-C performs better than HyperFlow-L in OOD tasks. This supports that the linear approximation of gradient flows oversimplifies the loss surface, which degrades the test-time adaptation ability. We provide further analysis on the effect of ODE objectives in Section 5.4.

Table 2: Few-shot classification accuracy (%) on CD-FSL benchmark. HyperFlow generalizes well on the out-of-domain tasks with various shots.

| Model Variant | Average | | | EuroSAT | | | CropDisease | | | ISIC | | | ChestX | | |
|---|---|---|---|---|---|---|---|---|---|---|---|---|---|---|---|
| | 5w5s | 5w20s | 5w50s | 5w5s | 5w20s | 5w50s | 5w5s | 5w20s | 5w50s | 5w5s | 5w20s | 5w50s | 5w5s | 5w20s | 5w50s |
| ViT-S backbone | | | | | | | | | | | | | | | |
| Direct Transfer | 56.53 | 61.95 | 63.88 | 80.56 | 86.45 | 87.90 | 34.89 | 42.74 | 45.93 | 85.80 | 90.44 | 91.46 | 24.88 | 28.15 | 30.23 |
| HyperFlow-L (Ours) | 61.07 | 66.64 | 67.70 | 83.45 | 88.49 | 89.60 | 46.74 | 55.66 | 56.24 | 89.19 | 92.61 | 94.22 | 24.89 | 29.78 | 30.73 |
| HyperFlow-C (Ours) | 62.82 | 67.24 | 69.60 | 86.92 | 91.25 | 91.91 | 46.16 | 53.31 | 57.79 | 93.05 | 95.30 | 96.23 | 25.14 | 29.09 | 32.47 |
| Head-Tuning | 57.46 | 63.53 | 65.96 | 80.42 | 85.88 | 88.93 | 37.17 | 46.29 | 50.52 | 87.46 | 92.56 | 93.52 | 24.79 | 29.37 | 30.86 |
| Bias-Tuning | 62.89 | 71.26 | 75.43 | 86.15 | 93.56 | 95.81 | 47.51 | 61.86 | 69.39 | 92.88 | 97.62 | 98.80 | 25.01 | 32.01 | 37.72 |
| LoRA-Tuning | 64.25 | 72.90 | 77.19 | 86.18 | 93.93 | 96.64 | 50.10 | 65.19 | 72.88 | 93.58 | 98.26 | 99.23 | 27.12 | 34.23 | 40.00 |
| Full-Tuning | 64.76 | 72.95 | 76.87 | 87.10 | 94.62 | 96.56 | 50.38 | 64.87 | 72.49 | 94.69 | 98.38 | 99.26 | 26.86 | 33.93 | 39.17 |
| ViT-L backbone | | | | | | | | | | | | | | | |
| Direct Transfer | 59.32 | 64.30 | 65.95 | 75.76 | 82.82 | 84.46 | 41.92 | 49.00 | 51.79 | 95.48 | 97.00 | 97.22 | 24.13 | 28.37 | 30.32 |
| HyperFlow-L (Ours) | 63.24 | 68.18 | 69.51 | 81.50 | 86.56 | 87.41 | 51.29 | 60.46 | 62.69 | 95.40 | 97.09 | 97.44 | 24.77 | 28.61 | 30.50 |
| HyperFlow-C (Ours) | 65.57 | 70.41 | 72.23 | 86.62 | 91.01 | 92.00 | 54.03 | 63.21 | 66.22 | 96.10 | 97.51 | 97.82 | 25.51 | 29.90 | 32.87 |
| Head-Tuning | 63.05 | 69.39 | 71.59 | 81.93 | 90.31 | 92.37 | 49.38 | 60.60 | 64.67 | 95.46 | 98.10 | 98.30 | 25.40 | 28.52 | 31.01 |
| Bias-Tuning | 65.37 | 72.88 | 76.47 | 86.44 | 94.73 | 96.79 | 52.97 | 66.36 | 73.01 | 96.20 | 98.96 | 99.39 | 25.88 | 31.48 | 36.68 |
| LoRA-Tuning | 65.95 | 73.72 | 77.77 | 87.37 | 94.86 | 97.01 | 53.40 | 67.80 | 75.31 | 96.61 | 98.94 | 99.48 | 26.43 | 33.26 | 39.27 |
| Full-Tuning | 66.23 | 73.61 | 78.15 | 87.87 | 95.25 | 97.28 | 53.08 | 67.86 | 75.10 | 97.26 | 99.14 | 99.60 | 26.72 | 32.20 | 40.63 |

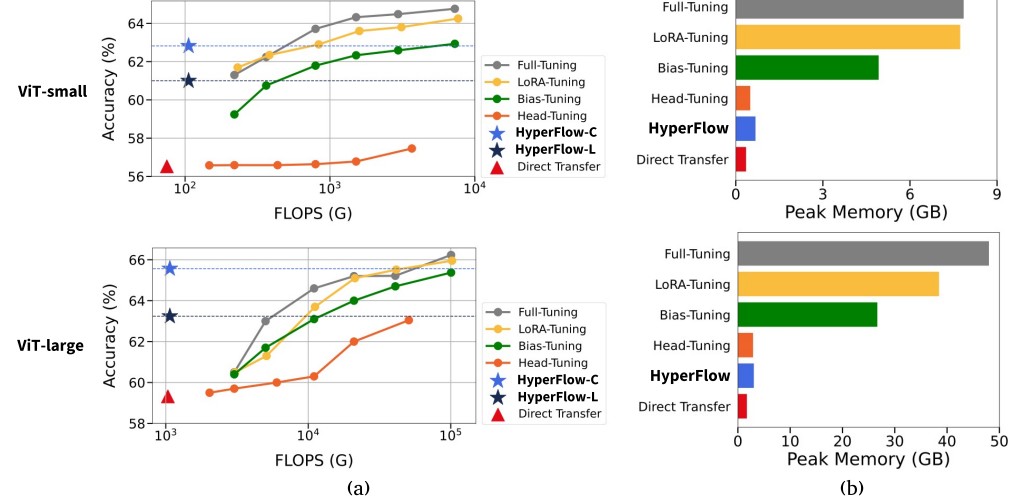

(a)       (b)

Figure 3: Computation cost analysis of HyperFlow. (a) Average 5-shot CD-FSL accuracy over FLOPs (plotted on a log scale). HyperFlow offers an effective middle ground between direct-transfer and fine-tuning. (b) Peak memory requirement in a 40-way 5-shot setting. HyperFlow introduces negligible memory overhead to Direct Transfer compared to Bias-Tuning and Full-Tuning.

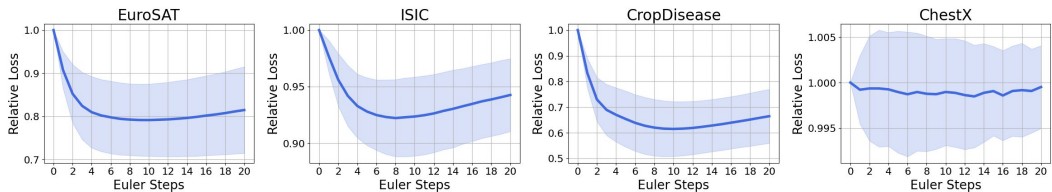

Figure 4: Relative losses on CD-FSL tasks during the ODE solving process of HyperFlow-C. We plot the average loss curves of 100 test episodes whose values are divided by the initial loss of each episode. The shaded region represents the range ± standard deviation. HyperFlow generalizes well to three unseen domains by consistently moving the parameters to the regions with lower loss values.

To further investigate the OOD generalization of HyperFlow, we plot the relative losses to the initial loss on CD-FSL 5-shot tasks during the ODE solving process of HyperFlow-C in Figure 4. In general, HyperFlow moves the parameters into regions with lower loss values, indicating that it effectively generalizes to OOD tasks. We observe that the loss does not drastically decrease on ChestX tasks, which can be attributed to the inherent limitation of the meta-trained ProtoNet backbone, since the fine-tuning baselines also show marginal improvement over Direct Transfer.

Table 3: Wall-clock time and storage required by each training stage of HyperFlow (ViT-S).

| | Meta-training | Offline trajectory collection | | | | | | Drift network training |
| | | HyperFlow-L | | | HyperFlow-C | | | |
| | | 400 traj | 4k traj | 40k traj | 400 traj | 4k traj | 40k traj | |
|---|---|---|---|---|---|---|---|---|
| Wall-clock time (hour) | 37.5 | 0.096 | 0.96 | 9.6 | 0.096 | 0.96 | 9.6 | 0.5 |
| Storage (GB) | 481 | 0.021 | 0.21 | 2.1 | 0.53 | 5.3 | 53 | - |

## 5.3 COMPUTATION COST ANALYSIS

In this section, we provide a detailed analysis of the memory and compute efficiency of HyperFlow. The comprehensive summary of the computation cost analysis is provided in Appendix E.

**Compute Efficiency** Figure 3-(a) shows the average accuracy of CD-FSL 5-shot tasks over FLOPs (FLoating point OPerations) of each method for a single test episode. By decoupling the adaptation process from the target model, HyperFlow requires merely $0.1\%$ of FLOPs of the Full-Tuning and Bias-Tuning approaches, translating to only $0.4\times$ increase over Direct Transfer. Therefore, Hyper-Flow occupies the middle ground between direct-transfer and fine-tuning in both performance and computation cost. It is noticeable that HyperFlow is even more efficient than the single-step Head-Tuning (the leftmost point on the orange line) while achieving much higher accuracy. These results suggest that HyperFlow can be highly advantageous in real-world end devices that demand cheap and fast computation for test-time adaptation.

**Memory Efficiency** Figure 3-(b) reports the peak memory usage of each method when processing a single 40-way, 5-shot episode with $128 \times 128$ images. We observe that HyperFlow reduces the memory requirement to just $6-14\%$ of that used by fine-tuning approaches, introducing an overhead of only $0.8-0.9\times$ memory compared to the Direct Transfer. The memory requirement of HyperFlow is even comparable to the simple Head-Tuning baseline, which only adapts the final classifier head. This substantial gain in memory efficiency stems from the fact that HyperFlow does not need to store the computation graph—its gradient-free adaptation runs entirely in inference mode. As a result, HyperFlow is particularly appealing for adapting large-scale models in low-resource environments, where fine-tuning would be infeasible due to the memory constraint.

**HyperFlow Training Overhead** Compared to the standard P>M>F pipeline, HyperFlow adds two offline stages: collecting fine-tuning trajectories and training the drift network. These stages incur extra time and storage overhead, but they are designed to be performed *once* on the server side, where larger computational resources are typically available, analogous to the pre-training and meta-training stages. Importantly, the overhead of HyperFlow training is modest relative to meta-training and can be made very small in practice. As summarized in Table 3, collecting 40k full trajectories of bias parameters and training the drift network introduces only about 25% additional training time and 11% additional storage compared to meta-training. Moreover, as we show in Section 5.4, HyperFlow can still achieve substantial gains over Direct Transfer with a much smaller trajectory budget (*e.g.*, 400 trajectories), under which the offline overhead becomes almost negligible.

## 5.4 ABLATION STUDY

In this section, we conduct ablation studies of HyperFlow with ViT-S backbone.

**Effect of ODE-based Inference** To investigate the impact of the ODE-based adaptation mechanism, we compare several variants of HyperFlow, including two non-ODE variants (**HyperNet** and **HyperFlow-D**), as well as the piecewise-linear variant (**HyperFlow-PL**) introduced in Section 3.3. All methods share the same training trajectories and architecture, but the non-ODE variants directly predict the next parameters rather than solving an ODE. Specifically, HyperNet predicts the final parameters $\theta_T$ given initial parameters $\theta_0$ in one step, while HyperFlow-D iteratively updates the parameters similar to the gradient descent. They can be seen as simplified versions of HyperFlow-L and HyperFlow-PL, where they only predict the drift at the discretized timesteps. Note that HyperNet is a natural generalization of the original hypernetworks proposed by Ha et al. (2017) for few-shot learning, using our implementation of the conditional drift network and target parameters.

Table 4 summarizes their performance on the Meta-Dataset and CD-FSL benchmarks. First, we observe that exploiting intermediate trajectories is beneficial in OOD generalization, where the non-linear variants (HyperFlow-D, HyperFlow-PL, and HyperFlow-C) consistently outperform the linear variants (HyperNet and HyperFlow-L). Also, HyperFlow-L outperforms HyperNet in CD-FSL tasks, which indicates that iterative updates at test-time adaptation can help OOD generalization. Finally, we observe that HyperFlow-C generally performs better than the non-smooth variants (HyperFlow-D and HyperFlow-C), which shows the robustness of the smooth ODE interpolation.

Table 4: Ablation study on the ODE objectives of HyperFlow.

| Model Variants | Meta-Dataset Average | | CD-FSL Average | | |
|---|---|---|---|---|---|
| | OOD | ID | 5w5s | 5w20s | 5w50s |
| HyperNet | 59.51 ± 1.01 | 83.56 ± 0.61 | 59.70 ± 0.49 | 64.93 ± 0.44 | 67.05 ± 0.41 |
| HyperFlow-L | 59.12 ± 1.01 | 83.42 ± 0.62 | 61.07 ± 0.53 | 66.64 ± 0.47 | 67.70 ± 0.43 |
| HyperFlow-PL | 61.59 ± 0.98 | 83.34 ± 0.63 | 62.62 ± 0.48 | 67.39 ± 0.41 | 69.60 ± 0.39 |
| HyperFlow-D | 61.47 ± 0.97 | 83.32 ± 0.63 | 62.19 ± 0.48 | 67.29 ± 0.41 | 69.23 ± 0.40 |
| HyperFlow-C | 63.11 ± 0.95 | 83.33 ± 0.63 | 62.82 ± 0.48 | 67.24 ± 0.41 | 69.60 ± 0.39 |

Table 5: Ablation study on the diversity of trajectories used to train HyperFlow.

| Diversity Factor | | Meta-Dataset Average | | CD-FSL Average | | |
|---|---|---|---|---|---|---|
| | | OOD | ID | 5w5s | 5w20s | 5w50s |
| # of Domains | 1 | 59.71 ± 1.02 | 83.34 ± 0.62 | 61.80 ± 0.49 | 67.21 ± 0.41 | 68.83 ± 0.41 |
| | 4 | 60.27 ± 1.02 | 83.34 ± 0.62 | 62.44 ± 0.48 | 67.55 ± 0.42 | 69.53 ± 0.40 |
| | 8 | 63.11 ± 0.95 | 83.33 ± 0.63 | 62.82 ± 0.48 | 67.24 ± 0.41 | 69.60 ± 0.39 |
| # of Episodes | 40 | 60.64 ± 0.99 | 83.32 ± 0.63 | 62.12 ± 0.49 | 67.39 ± 0.42 | 68.34 ± 0.42 |
| | 4000 | 63.11 ± 0.95 | 83.33 ± 0.63 | 62.82 ± 0.48 | 67.24 ± 0.41 | 69.60 ± 0.39 |
| # of Initializations | 1 | 61.76 ± 0.98 | 83.33 ± 0.63 | 62.67 ± 0.48 | 67.33 ± 0.41 | 69.73 ± 0.39 |
| | 10 | 63.11 ± 0.95 | 83.33 ± 0.63 | 62.82 ± 0.48 | 67.24 ± 0.41 | 69.60 ± 0.39 |

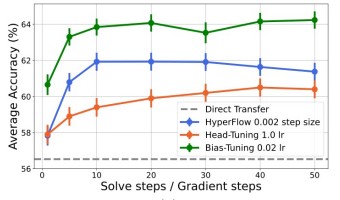 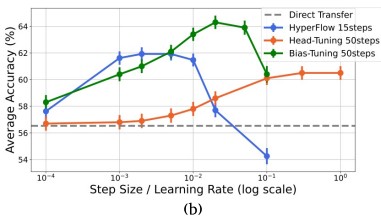

(a)  (b)

Figure 5: Hyper-parameter sensitivity of HyperFlow and fine-tuning baselines.

**Effect of Trajectory Diversity**  To further examine where the generalization ability of HyperFlow originates, we conduct ablation studies that vary the size and diversity of the training trajectories. Table 5 shows the results when HyperFlow-C is trained on different numbers of domains, tasks, and initializations. To analyze the effect of the number of domains, we train the model on a single domain (Inet) and four domains (Inet, Omglot, Acraft, CUB), respectively. Similarly, we train two additional variants that use only 40 episodes (five per domain) and a single initialization during training, respectively. As expected, performance increases consistently with larger and more diverse training trajectories, emphasizing that broader exploration of both the task space and the parameter space is crucial for robust generalization to unseen domains and tasks. However, surprisingly, even with the smallest amount of trajectories (40 episodes), HyperFlow still shows significant performance gains compared to Direct Transfer and Head-Tuning baselines, which can be attributed to the effective conditional drift architecture and the robust flow objective.

**Hyper-parameter Sensitivity**  During adaptation, HyperFlow has two hyper-parameters in the ODE solver: the step size and the number of solve steps. These respectively correspond to the learning rate and the number of gradient steps in the fine-tuning baselines, and we select them using a small validation set from each test dataset, following the learning-rate selection protocol of Hu et al. (2022b). To analyze sensitivity to these choices, we evaluate the performance of HyperFlow-C and the fine-tuning baselines on CD-FSL 5-shot tasks while varying either the step size (learning rate) or the number of steps. As shown in Figure 5, HyperFlow exhibits a stable performance region and consistently outperforms Direct Transfer and Head-Tuning across a broad range of step sizes and step counts, supporting that our gains do not rely on brittle hyper-parameter choices.

## 6 CONCLUSION

We introduced HyperFlow as an alternative to the two predominant few-shot learning paradigms: direct transfer and fine-tuning. By formulating the gradient descent procedure of fine-tuning as an ODE, we designed a conditional drift network that predicts the velocity field of the parameters given a few support examples. We focused updates solely on the bias parameters, enabling efficient training and inference on the parameter space. We also proposed approximate ODE objectives that continuously interpolate the discrete trajectories of gradient-descent simulations on the base dataset. Through the experiments, we demonstrated that HyperFlow significantly improves out-of-domain few-shot classification performance while adding minimal computational overhead to direct transfer, effectively occupying the middle ground between direct transfer and full fine-tuning.

**Reproducibility Statement** To help readers reproduce our experiments, we provided detailed descriptions of our architectures in Section 3.4 and Appendix C, and implementation details of the experiments in Section 5.1. We plan to release the source code to ensure the reproducibility of this paper.

**Ethics Statement** We have read the ICLR Code of Ethics and ensure that this work follows it. All data and pre-trained models used in our experiments are publicly available and have no ethical concerns.

**Use of LLMs** We used LLMs to check grammar and improve the clarity of the writing in this paper. We also used LLMs for finding the related work. The authors draft all the original content, and the role of the LLMs is strictly limited to language refinement. All final contents are carefully verified by the authors.

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
