# APPENDIX

## A ADDITIONAL RESULTS WITH CONVNEXT BACKBONE

In this section, we provide additional results with ConvNeXt-S (Liu et al., 2022) backbone to show the generality of HyperFlow to convolutional backbones without attention mechanisms. In this backbone, we choose the target parameters to be the bias parameters of convolution layers of the 5-th stage, which consists of 27 convolution layers, resulting in 10,368 parameters in total. Table 6 and 7 show the quantitative results of the ConvNeXt backbone, where Figure 6 shows the corresponding computation cost analysis. We observe that the overall trends observed from the ViT backbones are preserved, where HyperFlow establishes a practical middle ground between Direct Transfer and fine-tuning baselines. This result indicates that HyperFlow is robust to the choice of backbone and can be instantiated on both transformer and convolutional architectures.

| Model Variant | Average | | Out-of-Domain | | In-Domain | | | | | | | |
|---|---|---|---|---|---|---|---|---|---|---|---|---|
| | OOD | ID | Sign | COCO | Acraft | CUB | DTD | Fungi | Flower | Inet | Omglot | QDraw |
| Direct Transfer | 56.82 | 87.91 | 56.17 | 57.47 | 91.78 | 92.93 | 81.65 | 74.97 | 96.40 | 91.78 | 92.93 | 80.87 |
| HyperFlow-C (**Ours**) | 60.05 | 88.09 | 62.51 | 57.59 | 92.90 | 92.93 | 81.65 | 74.97 | 96.40 | 92.22 | 92.78 | 80.87 |
| Head-Tuning | 56.83 | 88.56 | 56.17 | 57.48 | 92.89 | 92.92 | 85.02 | 74.97 | 97.13 | 91.78 | 92.92 | 80.87 |
| Bias-Tuning | 73.35 | 88.85 | 86.78 | 59.92 | 92.99 | 93.33 | 86.05 | 74.98 | 97.10 | 92.06 | 93.05 | 81.26 |
| Full-Tuning | 74.53 | 89.27 | 90.19 | 58.87 | 93.84 | 93.35 | 88.46 | 74.87 | 97.18 | 92.03 | 93.18 | 81.27 |

Table 6: Few-shot classification accuracy (%) on Meta-Dataset benchmark with ConvNeXt-S backbone.

| Model Variant | Average | | | EuroSAT | | | CropDisease | | | ISIC | | | ChestX | | |
|---|---|---|---|---|---|---|---|---|---|---|---|---|---|---|---|
| | 5w5s | 5w20s | 5w50s | 5w5s | 5w20s | 5w50s | 5w5s | 5w20s | 5w50s | 5w5s | 5w20s | 5w50s | 5w5s | 5w20s | 5w50s |
| Direct Transfer | 57.08 | 63.00 | 65.46 | 77.58 | 85.32 | 87.01 | 37.82 | 46.32 | 50.35 | 87.89 | 91.42 | 92.55 | 25.04 | 28.92 | 31.91 |
| HyperFlow-C (**Ours**) | 60.08 | 64.60 | 66.54 | 84.84 | 86.81 | 87.61 | 41.60 | 50.33 | 53.57 | 88.83 | 92.33 | 93.30 | 25.04 | 28.91 | 31.67 |
| Head-Tuning | 60.98 | 67.98 | 71.09 | 82.97 | 90.24 | 92.58 | 42.99 | 55.35 | 60.79 | 92.17 | 96.14 | 97.2 | 25.77 | 30.19 | 33.77 |
| Bias-Tuning | 62.41 | 71.29 | 75.75 | 85.26 | 93.29 | 95.81 | 45.81 | 61.98 | 70.53 | 92.74 | 97.72 | 98.79 | 25.82 | 32.16 | 37.87 |
| Full-Tuning | 64.47 | 73.19 | 77.46 | 86.13 | 94.30 | 96.58 | 49.97 | 65.39 | 73.50 | 94.62 | 98.49 | 99.26 | 27.17 | 34.58 | 40.50 |

Table 7: Few-shot classification accuracy (%) on CD-FSL benchmark with ConvNeXt-S backbone.

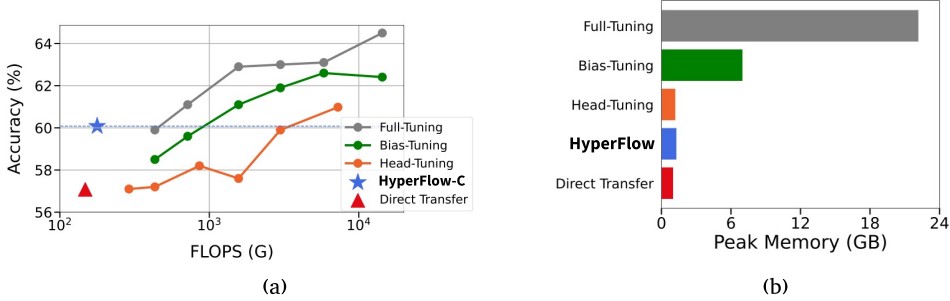

(a)                                                                  (b)

Figure 6: Computation cost analysis with ConvNeXt-S backbone. (a) Average 5-shot CD-FSL accuracy over FLOPs (plotted on a log scale). HyperFlow offers an effective middle ground between direct-transfer and fine-tuning. (b) Peak memory requirement in a 40-way 5-shot setting.

## B DETAILS ABOUT CUBIC FLOW

In this section, we explain the cubic flow objective introduced in Section 3.3 in detail. From the gradient simulation, each trajectory consists of $T$-length sequence of points $\{\theta_k\}_{k=1}^T$. The objective of cubic flow is to apply cubic splines on the points by treating $\theta(t)$ as a function of time where $\theta(k) = \theta_k, \ \forall k = 1, \cdots, T$. Specifically, after dividing the time interval $[0, T]$ into $T$ segments, we interpolate each segment $[k-1, k]$ with a cubic Hermite spline (Hermite, 1863; De Boor, 1978) that requires the two endpoints $\theta_{k-1}, \theta_k$ and their tangents $m_{k-1}, m_k$ to compute the coefficients

| hyperparameter | ViT-S | ViT-L | ConvNeXt-S |
|---|---|---|---|
| # of bias parameters ($n$) | 13,824 | 49,152 | 10,368 |
| hidden dimension ($d$) | 384 | 384 | 384 |
| # of conditional MLP blocks ($L$) | 1 | 1 | 1 |

Table 8: Architectural configuration of the dirft decoder of HyperFlow.

| Backbone | ViT-S | ViT-L | ConvNeXt-Small |
|---|---|---|---|
| Learning rate | 0.05 | 0.02 | 0.002 |
| Noise level ($\sigma$) | 0.2 | 0.2 | 0.015 |

Table 9: Learning rates and noise levels used in offline trajectory collection.

of the cubic curve. Since we uniformly divided the time interval, we can employ the Catmull-Rom spline (Catmull & Rom, 1974) that sets the tangents using two additional points $\theta_{k-2}, \theta_{k+1}$. Then, the point $p_t$ and its drift $v_t$ in the segment $[k-1, k]$ can be computed as follows:

$$\theta_t = a_k t^3 + b_k t^2 + c_k t + d_k, \tag{11}$$

$$v_t = 3a_k t^2 + 2b_k t + c_k, \tag{12}$$

where the coefficients $a_k, b_k, c_k, d_k$ are computed by the four points $\theta_{k-2}, \theta_{k-1}, \theta_k, \theta_{k+1}$. as follows:

$$a_k = -\frac{1}{2}\theta_{k-2} + \frac{3}{2}\theta_{k-1} - \frac{3}{2}\theta_k + \frac{1}{2}\theta_{k+1} \tag{13}$$

$$b_k = \theta_{k-2} - \frac{5}{2}\theta_i + 2\theta_k - \frac{1}{2}\theta_{k+1} \tag{14}$$

$$c_k = -\frac{1}{2}\theta_{k-2} + \frac{1}{2}\theta_k \tag{15}$$

$$d_k = \theta_{k-1} \tag{16}$$

For interpolating the first and the last segment, we set the sentinels as $\theta_{-1} = \theta_0$ and $\theta_{T+1} = \theta_T$.

## C  DETAILS ABOUT THE DRIFT DECODER

In this section, we provide detailed explanations about the drift decoder and its architectural configuration.

As introduced in Section 3.4, the drift decoder is a shallow MLP whose intermediate features are modulated by adaptive layer normalization (Peebles & Xie, 2023) (AdaLN) modules. The drift decoder consists of a linear input layer, $L$ conditional MLP blocks, and a linear output layer. First, the input layer transforms the flattened parameters $\theta_t \in \mathbb{R}^n$ into a latent vector $\mathbf{p}^{(0)} \in \mathbb{R}^d$, where $n$ and $d$ are the number of bias parameters and the hidden dimension, respectively. Then, each conditional MLP block at layer $l$ gets the latent features from the previous layer $\mathbf{p}^{(l-1)}$ and a condition vector $\mathbf{c}^{(l)}$, which is the concatenation of the task vector $z_\mathcal{T}$ from support encoder and the time embeding vector, to produce output features $\mathbf{p}^{(l)}$ as follows:

$$\mathbf{p}^{(l)} = \mathbf{p}^{(l-1)} + \alpha(\mathbf{c}^{(l)}) \cdot MLP(\gamma(\mathbf{c}^{(l)}) \cdot LN(\mathbf{p}^{(l-1)}) + \beta(\mathbf{c}^{(l)})), \tag{17}$$

where $MLP$ is a 2-layer MLP and $\alpha, \beta, \gamma$ are the AdaLN layers that produce gating, shift, and scaling parameters, respectively. Each AdaLN layer consists of the SiLU activation (Elfwing et al., 2018) and a linear layer. Finally, the output layer transforms the last latent features $p^{(L)} \in \mathbb{R}^d$ into the drift $v_t \in \mathbb{R}^n$.

In Table 8, we provide the architectural configuration of the drift decoder used in our experiments. All the variants (HyperNet, HyperFlow-L, HyperFlow-C, HyperFlow-D, and HyperFlow-PL) follow the same architecture.

# D  DETAILS ABOUT THE TRAJECTORY COLLECTION

During the offline trajectory collection, we use Adam Kingma & Ba (2015) optimizer to fine-tune the bias parameters of the base model. Similar to the episodic meta-training, we randomly sample classification tasks from the base dataset and fine-tune the parameters with these tasks. Also, we add a random perturbation $\epsilon \sim \mathcal{N}(0, \sigma^2)$ to the initialization of the parameters. Table 9 summarizes the learning rates and initial noise used in the offline trajectory collection of various backbones.

# E  COMPREHENSIVE COMPUTATION COST ANALYSIS

In Table 10-12, we provide the comprehensive analysis on the computation cost of the baselines in each of the three backbones, including FLOPs, peak memory, and the number of parameters.

| Model | Euler/Adam steps | FLOPs (G) | Peak Memory (MB) | Parameters (M) |
|---|---|---|---|---|
| Direct Transfer | - | 74.89 | 361.62 | 21.67 |
| HyperFlow | 1 | 104.59 | 694.79 | 45.00 |
|  | 10 | 104.81 |  |  |
|  | 50 | 105.76 |  |  |
| Head-Tuning | 1 | 146.9 | 510.04 | 21.68 |
|  | 2 | 218.91 |  |  |
|  | 5 | 434.93 |  |  |
|  | 10 | 794.98 |  |  |
|  | 20 | 1,515.06 |  |  |
|  | 50 | 3,675.32 |  |  |
| Bias-Tuning | 1 | 218.91 | 5,039.52 | 21.67 |
|  | 2 | 362.92 |  |  |
|  | 5 | 794.98 |  |  |
|  | 10 | 1,515.06 |  |  |
|  | 20 | 2,955.24 |  |  |
|  | 50 | 7,275.76 |  |  |
| LoRA-Tuning | 1 | 230.56 | 7,916.21 | 22.85 |
|  | 2 | 382.25 |  |  |
|  | 5 | 837.30 |  |  |
|  | 10 | 1,595.73 |  |  |
|  | 20 | 3,112.58 |  |  |
|  | 50 | 7,663.13 |  |  |
| Full-Tuning | 1 | 218.91 | 8,038.23 | 21.67 |
|  | 2 | 362.92 |  |  |
|  | 5 | 794.98 |  |  |
|  | 10 | 1,515.06 |  |  |
|  | 20 | 2,955.24 |  |  |
|  | 50 | 7,275.76 |  |  |

Table 10: Comprehensive analysis of the computation cost required by the adaptation methods on ViT-S backbone.

| Model | Euler/Adam steps | FLOPs (G) | Peak Memory (MB) | Parameters (M) |
|---|---|---|---|---|
| Direct Transfer | - | 1,034.73 | 1,765.11 | 303.21 |
| HyperFlow | 1 | 1,064.48 | 3,125.52 | 353.71 |
| | 10 | 1,065.18 | | |
| | 50 | 1,068.31 | | |
| Head-Tuning | 1 | 2,029.65 | 2,970.66 | 303.22 |
| | 2 | 3,024.58 | | |
| | 5 | 6,009.37 | | |
| | 10 | 10,984.01 | | |
| | 20 | 20,933.29 | | |
| | 50 | 50,781.14 | | |
| Bias-Tuning | 1 | 3,024.58 | 27,307.70 | 303.21 |
| | 2 | 5,014.44 | | |
| | 5 | 10,984.01 | | |
| | 10 | 20,933.29 | | |
| | 20 | 40,831.85 | | |
| | 50 | 100,527.55 | | |
| LoRA-Tuning | 1 | 3,071.20 | 39,385.78 | 309.50 |
| | 2 | 5,091.73 | | |
| | 5 | 11,153.31 | | |
| | 10 | 21,255.95 | | |
| | 20 | 41,461.22 | | |
| | 50 | 102,077.04 | | |
| Full-Tuning | 1 | 3,024.58 | 49,156.63 | 303.21 |
| | 2 | 5,014.44 | | |
| | 5 | 10,984.01 | | |
| | 10 | 20,933.29 | | |
| | 20 | 40,831.85 | | |
| | 50 | 100,527.55 | | |

Table 11: Comprehensive analysis of the computation cost on ViT-L backbone.

| Model | Euler/Adam steps | FLOPs (G) | Peak Memory (MB) | Parameters (M) |
|---|---|---|---|---|
| Direct Transfer | - | 147.77 | 1,026.04 | 49.45 |
| HyperFlow | 1 | 177.45 | 1,319.35 | 70.13 |
| | 10 | 177.62 | | |
| | 50 | 178.36 | | |
| Head-Tuning | 1 | 289.86 | 1,224.62 | 49.48 |
| | 2 | 431.94 | | |
| | 5 | 858.21 | | |
| | 10 | 1,568.64 | | |
| | 20 | 2,989.51 | | |
| | 50 | 7,252.12 | | |
| Bias-Tuning | 1 | 431.94 | 7,159.56 | 49.45 |
| | 2 | 716.12 | | |
| | 5 | 1,568.64 | | |
| | 10 | 2,989.51 | | |
| | 20 | 5,831.25 | | |
| | 50 | 14,356.47 | | |
| Full-Tuning | 1 | 431.94 | 22,708.68 | 49.45 |
| | 2 | 716.12 | | |
| | 5 | 1,568.64 | | |
| | 10 | 2,989.51 | | |
| | 20 | 5,831.25 | | |
| | 50 | 14,356.47 | | |

Table 12: Comprehensive analysis of the computation cost on ConvNeXt-S backbone.