# OpenReview forum: "HyperFlow: Gradient-Free Emulation of Few-Shot Fine-Tuning"
_ICLR.cc/2026/Conference — Submitted to ICLR 2026_

### Official Review · Reviewer_M6KB · 2025-10-30

**Soundness:** 1
**Presentation:** 2
**Contribution:** 2
**Rating:** 2
**Confidence:** 3

**Summary:**

The paper introduces HyperFlow, a test-time adaptation method for few-shot learning that emulates gradient descent without backpropagation. It models fine-tuning as an ODE (gradient flow) and trains a lightweight, task-conditioned drift network that takes a few-shot support set and predicts parameter updates. At inference, adaptation reduces to forward-only ODE steps, yielding strong efficiency: the authors report roughly 7% of peak memory and 0.1% of FLOPs compared to standard fine-tuning, while improving out-of-domain (OOD) accuracy over non-fine-tuned baselines. Despite compelling efficiency gains, the approach is weakened by its lack of offline cost and its inferior accuracy compared to standard (and even bias-only) fine-tuning.

**Strengths:**

- Casting fine-tuning as a learned task-conditioned ODE enables gradient-free adaptation using only forward passes.
- **Efficiency:** Clear compute/memory wins while beating direct transfer (and some head-tuning) on OOD benchmarks.

**Weaknesses:**

- **Unreported offline cost:** The method depends on generating fine-tuning trajectories and training the drift network; the compute, time, and storage for these steps are not quantified, making the total cost unclear.
- **Accuracy gap vs. stronger baselines:** Few-shot accuracies lag full fine-tuning and bias-only fine-tuning, which questions when HyperFlow is preferable beyond strict resource constraints.
- **Scalability:** The approach appears tied to a restricted parameter subset; it’s unclear how performance and efficiency trade off when expanding beyond that (e.g., LoRA/adapters) or switching backbones.

**Questions:**

- What are the FLOPs, wall-clock time, and storage required for (i) collecting fine-tuning trajectories and (ii) training the conditional drift network with interpolated ODEs?
- How sensitive is performance to the number/size of ODE steps at test time?
- How does accuracy/efficiency change when expanding the adapted parameter set (e.g., LoRA/adapters)?

---

> ### Author Response · Authors · 2025-11-28
>
> > **W1/Q1**. Unreported offline cost
>
> **A1**. We appreciate the reviewer’s request for a detailed breakdown of the offline computational and storage costs. We have added **Table 3** in the revised paper to quantify the wall-clock time and storage footprint. Our analysis confirms that these costs are modest relative to the meta-training stage and can be reduced to negligible levels if needed.
>
> As clarified above, the primary design goal of HyperFlow is to **minimize test-time adaptation cost** for deployment on resource-constrained devices. The offline trajectory collection is a **one-time server-side procedure**, analogous to the pre-training and meta-training stages in the standard PMF pipeline. In realistic deployment scenarios, this one-time offline investment enables the model to perform gradient-free adaptation for many user-specific tasks. Therefore, while we acknowledge the existence of an offline cost, the relevant metric for comparison is the **amortized per-episode cost**, under which HyperFlow provides a substantial efficiency gain over standard fine-tuning.
>
> Regarding the specific costs, we find them to be **modest** when compared to the meta-training stage. As shown in Table 3, the combined process of collecting 40k full trajectories and training the drift network introduces a computational overhead of approximately **25%** relative to the time already required for the backbone’s meta-training stage. Similarly, storing the trajectory vectors adds only about **11%** storage overhead compared to the size of the meta-training image dataset. This efficiency is enabled by our strategic choice to target the bias parameter subset, which significantly reduces the dimensionality of the trajectory data compared to full-weight matrices. **HyperFlow-L** does not require intermediate trajectory states and further reduces the storage overhead to roughly **0.4%**.
>
> Crucially, HyperFlow **does not strictly require 40k trajectories** to be effective. As we already showed in Table 4 of the Ablation Study section, the method still achieves significant performance gain over Direct Transfer with fewer trajectories. Reducing the dataset to just 400 trajectories lowers the computational overhead to **~2%** and the storage overhead to **~0.2%** relative to the meta-training stage. Despite this massive reduction in offline cost, HyperFlow maintains strong OOD performance (e.g., improving over Direct Transfer by **~5.5%** on Meta-Dataset and CD-FSL), effectively retaining the benefits of the method while eliminating the computational burden.
>
> > **W2.** Accuracy gap vs. stronger baselines
>
> **A2**. We agree that full fine-tuning and bias-tuning achieve higher raw accuracy when their gradient cost is affordable. However, our goal with HyperFlow is to optimize the **accuracy–efficiency trade-off**, not to dominate unconstrained fine-tuning in all settings.
> Full fine-tuning and bias-tuning rely on backpropagation through the backbone, requiring substantial peak memory (to store activations) and computation. On many target edge devices (e.g., IoT, mobile, robotics), such methods are often impractical or impossible due to hardware constraints. In these scenarios, gradient-based fine-tuning serves as a **reference upper bound**, rather than a directly deployable competitor.
>
> Against practical deployable baselines (Direct Transfer, linear head-tuning), HyperFlow consistently improves OOD accuracy. As shown in Figure 3, HyperFlow achieves a distinct operating point: it recovers a substantial fraction of the fine-tuning gains while requiring only **6–14%** of the peak memory and approximately **0.1%** of the FLOPs of gradient-based methods. From this perspective, the goal of HyperFlow is not to match full fine-tuning in unconstrained settings, but to enable effective adaptation in regimes where gradient-based updates are constrained or unavailable.

---

> > ### Author Response · Authors · 2025-11-28
> >
> > > **W3/Q3**. Scalability / Expanding the adapted parameter set (e.g., LoRA/adapters)
> >
> > **A3**. We agree that the choice of the adapted parameter subset is central to the scalability and efficiency of HyperFlow. In our experiments, we found that targeting the **bias subset** offers a particularly favorable trade-off in the few-shot regime.
> >
> > While we agree that LoRA is powerful, our new experiments (updated Tables 1–2) show that, in the few-shot cross-domain setting we study, bias-tuning already captures most of the adaptation gains. Extending HyperFlow to emulate LoRA trajectories would substantially increase the dimensionality of the drift network’s output (and thus its inference cost) for only marginal accuracy gains over bias-based HyperFlow in our setting. We therefore chose the bias subset to maximize the **efficiency-to-accuracy ratio** in this regime.
> >
> > To address concerns about backbones, we also applied HyperFlow to **ViT-L** (14× larger than ViT-S) and **ConvNeXt-S**. As shown in **Tables 1–2 and 6–7**, HyperFlow scales effectively to these modern architectures, maintaining the same relative efficiency gains and qualitative behavior. This confirms that the method is not tied to a specific backbone or to very small models.
> >
> > > **Q2**. Sensitivity to number/size of ODE steps at test time?
> >
> > **A4**. We have included a **sensitivity analysis in Figure 5** in the revised paper, plotting test accuracy as a function of either step size (learning rate in case of fine-tuning) or number of steps. These plots show that HyperFlow maintains a **stable performance region** and consistently outperforms Direct Transfer and head-tuning across a broad range of step sizes, indicating that our gains are not due to brittle hyperparameter choices.

---

### Official Review · Reviewer_Vcrg · 2025-10-31

**Soundness:** 3
**Presentation:** 2
**Contribution:** 3
**Rating:** 6
**Confidence:** 3

**Summary:**

This paper reinterprets test-time few-shot adaptation as the gradient-flow ODE and trains a lightweight, support-set–conditioned drift network to predict the parameter velocity field, enabling adaptation without backpropagation via only a few steps of Euler integration. On Meta-Dataset and CD-FSL, HyperFlow improves over Direct Transfer while requiring only about 0.1% of the FLOPs and 7–10% of the peak memory of standard finetuning, thereby achieving a practical middle ground in the performance–cost trade-off.

**Strengths:**

1. Intuitive to understand
Framed as “fine-tuning = Euler discretization of a gradient-flow ODE” the pipeline is explained clearly: a task representation from the support set conditions a drift network whose forward passes, combined with simple numerical integration, effect adaptation without gradients. The interpolation methods—linear, piecewise-linear, and cubic —are well organized and accessible.

2. Systematic Experimental Design
Fairness Controls: Comparisons include Direct Transfer, linear-head, bias-tuning, and full fine-tuning. To ensure a fair budget, HyperFlow-L/C performs up to 50 Euler steps, aligning the number of update steps with gradient-based fine-tuning.
By presenting results that vary the ODE objectives and the diversity of training trajectories (domains, episode counts, and initialization noise), the paper provides a factor-wise analysis of performance and generalization.

**Weaknesses:**

1. Backbone Generalization
The classifier is ProtoNet + ViT-Small, and the drift network’s task encoder is a frozen ResNet-18. The paper provides no empirical evaluation on alternative backbones (e.g.CLIP-ViT), leaving backbone generalization as an open limitation.

2. Implicit Gradient Dependency and Limited Generalization
While HyperFlow claims gradient-free adaptation, its drift network is trained on gradient trajectories from base tasks, meaning gradient information is still implicitly used. This raises concerns about how well the learned dynamics generalize when the loss landscape or data distribution differs from training, and since training the drift network itself requires computing gradients over simulated trajectories, the claimed efficiency advantage should be reconsidered in light of this additional training cost.(If the drift network requires additional training for distribution shifts)

3. Missing theoretical guarantees for the interpolation-based ODE objective
Relies on linear/PL/cubic interpolation and a learned drift but provides no formal guarantees on convergence, Euler-step stability, or drift–gradient approximation error. Potential issues aren’t analyzed, and no sufficient conditions are given—evidence is empirical only.

4. Under-reported offline trajectory cost
The paper does not quantify the compute/energy/storage cost of collecting trajectories as a function of episodes, initializations, and GD steps, nor provide budget–accuracy trade-offs or a break-even analysis versus fine-tuning.

**Questions:**

1. Would the performance gap from direct transfer still remain if the authors applied it to CLIP-ViT(B/16, L/14), or might it actually vanish?
2. If the authors tried a model like ConvNeXt, which lacks the qkv mechanism, how could this method be adapted — and would it still lead to any measurable performance improvement?
3. On the ChestX dataset, the loss decreased only gradually — is that truly due to the backbone’s limitation, or could a different backbone reveal a different trend?
4. It seems that the results in Figure 3(a) and Tables 1 and 2 are inconsistent. Specifically, the performance of HyperFlow-C in Figure 3(a) appears to be better than that of Bias-Tuning, whereas in the tables it is not. Could you clarify why there is a discrepancy between these results?

---

> ### Author Response · Authors · 2025-11-28
>
> > **W1/Q1-3**. Backbone Generalization / Application to CLIP-ViT(B/16, L/14) and ConvNeXt / Behavior of the ChestX dataset in other backbones.
>
> We agree that it is important to test HyperFlow beyond a single ViT-Small backbone. To address **W1 and Q2**, we have added experiments with two additional encoders under the same PMF ProtoNet pipeline: **ViT-L** and **ConvNeXt-S** (which has no qkv mechanism). For each backbone, we select a small PEFT subset analogous to the qkv-bias subset used for ViT-S (e.g., biases in key projection / convolutional blocks) and re-train HyperFlow using the corresponding trajectories. As summarized in **Table 1,2,6,7** and **Figure 3,6** in the revised paper, we observe the same qualitative pattern for both ViT-L and ConvNeXt-S as for ViT-S: (1) Direct transfer performs worst on cross-domain tasks, (2) Gradient-based fine-tuning (bias/full) achieves the best accuracy but with high test-time FLOPs and memory, and (3) HyperFlow consistently sits between these two, recovering a substantial fraction of the fine-tuning gains while retaining the forward-only, low-footprint adaptation. These results indicate that HyperFlow is robust to the choice of backbone and can be instantiated on both transformer and convolutional architectures.
>
> Regarding **ChestX (Q3)**, we find that the same behavior holds across all three backbones (ViT-S, ConvNeXt-S, ViT-L): the loss decreases only gradually and the absolute gains remain limited even with full fine-tuning. HyperFlow again improves over direct transfer but with a smaller margin than on other OOD datasets. Since this pattern is consistent across very different encoders, it suggests that the main bottleneck on ChestX is the large domain gap between the meta-training data (natural images) and the test data (medical X-rays), rather than a failure mode specific to HyperFlow.
>
> Finally, for **CLIP-ViT (Q1)**, while we agree CLIP is valuable, few-shot adaptation for vision-language models involves distinct challenges (prompt engineering, text-branch tuning) that are orthogonal to the PMF-ProtoNet computer vision setting we target. By focusing on standard vision backbones (ViT-S, ViT-L, ConvNeXt-S), we isolate the contribution of our method—gradient-free dynamics learning—without conflating it with prompt tuning strategies. We therefore leave HyperFlow+CLIP as future work; the new results on ViT-L and ConvNeXt-S support that HyperFlow is **not tied to a specific backbone**, and we expect the qualitative pattern “direct transfer < HyperFlow < fine-tuning” to persist under other reasonable backbone choices, though we do not claim this as a proven result for CLIP in the current submission.

---

> > ### Author Response · Authors · 2025-11-28
> >
> > > **W2**. Implicit Gradient Dependency and Limited Generalization
> >
> > **A2**. We agree that HyperFlow is not gradient-free during training: the drift network is trained on gradient-based fine-tuning trajectories collected on meta-train tasks. Our claim of “gradient-free adaptation” is strictly about **test time** on novel tasks within the same meta-learning setting: once the drift is trained, adaptation requires only forward passes through the base model and the drift network, with no backpropagation or gradient computation on the target episodes.
> >
> > Conceptually, HyperFlow should be viewed as a meta-learning / hypernetwork–style optimizer, rather than as a universally applicable replacement for gradients. Just like ProtoNet and other cross-domain few-shot meta-learners, we train HyperFlow on a diverse set of meta-train tasks (Meta-Dataset / CD-FSL sources), and then apply it once to all in-domain and OOD test tasks in these benchmarks without any retraining. In this regime, we do not claim that HyperFlow is “more robust” to distribution shift than the underlying meta-trained ProtoNet. Rather, its ability to generalize to the loss landscapes of unseen tasks—up to a certain extent—can be attributed to two factors:
> >
> > 1. **Task diversity in meta-training**. The drift network is exposed to many source domains and tasks during trajectory collection and is optimized to produce an update rule that works well on average over this meta-training distribution, just as the ProtoNet backbone itself is meta-trained to produce features that transfer across domains.
> >
> > 2. **Amortized learning of shared dynamics**. The drift network is trained in an amortized fashion across tasks and timesteps, which encourages it to capture shared patterns in how parameters are updated across tasks, rather than memorizing individual trajectories. This shared optimizer can then transfer to new tasks that are “similar enough” to those seen in meta-training, in the same sense that a meta-trained ProtoNet transfers to new OOD episodes in Meta-Dataset and CD-FSL.
> >
> > If one moves to a substantially different regime (e.g., a very different data family or a new backbone), retraining the drift network—analogous to re-running meta-training for ProtoNet or other meta-learners—would indeed be necessary, and that offline cost should be accounted for in that new setting. This is a common limitation of meta-learning and hypernetwork approaches, and HyperFlow shares it rather than eliminating it. In the reported experiments, however, no such retraining is performed: we train **one drift network per backbone** on the prescribed meta-train domains and use it unchanged for all in-domain and OOD test domains in Meta-Dataset and CD-FSL, while benefiting from gradient-free, efficient adaptation at test time.

---

> > > ### Author Response · Authors · 2025-11-28
> > >
> > > > **W4**. Under-reported offline trajectory cost
> > >
> > > **A3**. We appreciate the reviewer’s request for a detailed breakdown of the offline computational and storage costs. We have added **Table 3** in the revised paper to strictly quantify the wall-clock time and storage footprint. Our analysis confirms that these costs are modest relative to the meta-training stage and can be reduced to negligible levels if needed.
> > >
> > > We first clarify that the primary design goal of HyperFlow is to **minimize test-time adaptation cost** for deployment on resource-constrained devices. The offline trajectory collection is a **one-time server-side procedure**, analogous to the pre-training and meta-training stages in the standard PMF pipeline. In realistic deployment scenarios, this one-time offline investment enables the model to perform gradient-free adaptation for millions of user-specific tasks. Therefore, while we acknowledge the existence of an offline cost, the relevant metric for comparison is the **amortized per-episode cost**, under which HyperFlow provides a massive efficiency gain over standard fine-tuning.
> > >
> > > Regarding the specific costs, we find them to be **modest** when compared to the meta-training stage. As shown in Table 3, the combined process of collecting 40k full trajectories and training the drift network introduces a computational overhead of approximately **25%** relative to the time already required for the backbone's meta-training stage. Similarly, storing the trajectory vectors adds only **~11%** storage overhead compared to the size of the meta-training image dataset. This efficiency is enabled by our strategic choice to target the bias parameter subset, which significantly reduces the dimensionality of the trajectory data compared to full-weight matrices. Also note that **HyperFlow-L** does not require intermediate trajectories and further reduces the storage overhead to **~0.4%**.
> > >
> > > Crucially, HyperFlow **does not strictly require 40k trajectories** to be effective. As we already showed in Table 4 of the Ablation Study section, the method still achieves significant performance gain over Direct Transfer with fewer trajectories. Reducing the dataset to just 400 trajectories lowers the computational overhead to **~2%** and the storage overhead to **~0.2%** relative to the meta-training stage. Despite this massive reduction in offline cost, HyperFlow maintains strong OOD performance (e.g., improving over Direct Transfer by **~5.5%** on Meta-Dataset and CD-FSL), effectively retaining the benefits of the method while eliminating the computational burden.
> > >
> > > > **W3.** Missing theoretical guarantees for the interpolation-based ODE objective
> > >
> > > **A4**. We agree that our analysis is primarily empirical, and we do not claim formal convergence guarantees or tight bounds on drift–gradient approximation error. The interpolation choices (linear, piecewise-linear, cubic) are used to construct continuous-time surrogates of the discrete gradient trajectories, from which we derive training targets for the drift. In this work we do not attempt a full theoretical analysis of these objectives in the non-convex, high-dimensional setting of deep networks.
> > >
> > > Instead, we focus on a systematic empirical evaluation that directly probes the relevant issues. First, we ablate across linear / PL / cubic objectives (Table 3), showing that cubic interpolation most closely tracks the behavior of gradient descent and yields the best overall performance in both Meta-Dataset and CD-FSL. Second, we plot loss curves over Euler steps for HyperFlow (Figure 4), where we do not observe catastrophic divergence or instability except for ChestX. Finally, we study how performance changes as we vary the diversity and number of trajectories used for training (Table 4), and observe that the increased diversity of training trajectories improves the generalization ability of HyperFlow.
> > >
> > > > **Q4**. Figure 3(a) and Tables 1 and 2 are inconsistent.
> > >
> > > **A5**.  We thank the reviewer for carefully checking Figure 3(a) against Tables 1 and 2. We re-examined the underlying numbers and confirmed that the plotted values of HyperFlow-C and Bias-Tuning in Figure 3(a) match the table entries exactly. The apparent discrepancy arises from **visual ambiguity**: the combination of marker sizes, sampling of x-axis points, and the y-axis scale makes HyperFlow-C’s curve appear slightly above Bias-Tuning in certain regions, even though the actual values are slightly lower.
> > >
> > > To avoid this confusion, in the revised paper, we re-plotted Figure 3(a) with smaller markers and clearer grid lines, and added an auxiliary horizontal line at the HyperFlow-C point. This makes it clear that Figure 3(a) and Tables 1–2 are fully consistent and that HyperFlow-C does not surpass Bias-Tuning on the reported averages.

---

### Official Review · Reviewer_6hQt · 2025-11-01

**Soundness:** 3
**Presentation:** 3
**Contribution:** 3
**Rating:** 4
**Confidence:** 4

**Summary:**

The paper proposes HyperFlow, a method for gradient-free few-shot learning. A hypernetwork is trained based on few-shot training trajectories. Given a conditioning signal (the shots) and a continuous update step, this network learns to predict the gradient of the model parameters. When new shots are presented to the hypernetwork, an integration over the simulated gradient updates can be carried out to adapt a base model.

**Strengths:**

The paper is well written and easy to follow. The method is explained well and all experimental setups are clear. While the presented performance gains are marginal compared to bias-tuning, the method greatly reduced the memory footprint and computational cost of finetuning.

**Weaknesses:**

The experimental part is limited, comparing HyperFlow to 1) direct transfer, 2) head-tuning, 3) bias-tuning and 4) full-tuning. However, it is known that parameter efficient finetuning methods like LoRA (https://arxiv.org/abs/2106.09685) or sparse variants are very powerful for few-shot adaptation of pre-trained methods.

Extending the experiments could clearly help to strenghten the paper’s contribution.

**Questions:**

I have a few conceptual questions:
1) Does the hyper-network generalize to increasing/decreasing number of shots?
2) Is the ODE solver always executed for a fixed number of steps? Does the gradient estimated by the hyper-network vanish after these steps?
3) For a linear flow as proposed in Eq. 5-6, the time derivative is constant. Hence, integration boils down to a simple multiplication with the time interval. So effectively, the hyper-network is trained to predict a scaled one-step update. Is this correct?

About the results:
1) In Table 2, the HyperFlow method sometimes improves upon bias-tuning. For me this is a bit strange, since HyperFlow is trained to approximate bias-tuning episodes. I would love to hear some thoughts about that.
2) Figure 4 shows the relative loss over the Euler steps. It seems that the update is not stable, because the loss increases in the end. Is this due to overfitting or because the predicted gradient does not vanish and we overshoot the optimal point?
3) Another question related to stability: Did you try if running Euler integration on training data could recover the optima from the training episodes?

And finally: Am I right that Eq.6 should be the time derivative of Eq.5?

---

> ### Comment · Reviewer_6hQt · 2025-11-26
>
> Is there any interest in discussion?

---

> > ### Author Response · Authors · 2025-11-28
> >
> > > **W1**. Additional baselines like LoRA.
> >
> > **A1**. We appreciate the reviewer’s suggestion to compare with advanced parameter-efficient fine-tuning (PEFT) methods such as LoRA. In response, we have added a **LoRA-tuning** baseline to our experiments. The results are included in **Tables 1–2** and **Figure 3** in the revised paper.
> >
> > LoRA generally shows a **marginal improvement** over bias-tuning in our few-shot setting, but it exposes substantially more trainable parameters than bias-tuning and still requires full gradient backpropagation through the backbone. As a result, it incurs significantly higher test-time memory and compute (about **11–13× peak memory** and **70–100× FLOPs** compared to HyperFlow). Thus, while LoRA is indeed a strong PEFT method in terms of accuracy, HyperFlow targets a **different operating point**: scenarios where gradient-based adaptation is expensive or infeasible, and one needs a forward-only adaptation mechanism with very low memory and compute. In these resource-constrained settings, HyperFlow offers clear efficiency benefits over gradient-based methods such as LoRA, while still improving significantly over non-adaptive baselines.
> >
> >
> > > **Q1**. Effect of increasing/decreasing the number of shots
> >
> > **A2**. Yes, the drift network generalizes across different numbers of shots. The support set is encoded via a **permutation-invariant pooling** of per-example features, so the drift network can in principle handle an arbitrary number of support examples.
> >
> > Empirically, we already evaluated HyperFlow with **5, 20, and 50 shots** on CD-FSL (**Table 2** in the main paper). As the number of shots increases, HyperFlow’s accuracy improves consistently, and it remains between the naive baselines (direct transfer, head-tuning) and gradient-based fine-tuning across all shot configurations. This indicates that the drift network leverages additional support examples in a stable way, rather than being tied to a fixed shot count.
> >
> >
> > > **Q2**. Is the ODE solver always executed for a fixed number of steps? Does the gradient estimated by the hyper-network vanish after these steps?
> >
> > **A3**. For each dataset, we select a **single fixed number of Euler steps** using a small validation set, as described in the paper. Concretely, for each candidate step size we integrate up to 50 steps and choose the step count that maximizes validation accuracy; we then fix this pair (step_size, n_steps) for **all** test tasks on that dataset. Thus, at test time the ODE solver is always run for a fixed number of steps per dataset; it is not run until the predicted drift vanishes. Instead, we stop at the step that gives the best validation performance, analogous to choosing the number of gradient steps in standard fine-tuning.
> >
> > This protocol mirrors the learning-rate search used for fine-tuning methods in Hu et al. (2023): all methods (bias-tuning, full-tuning, and HyperFlow) use the same small set of validation episodes and a simple grid over their respective step-size parameters (learning rate for fine-tuning, Euler step size for HyperFlow), with no per-task tuning at test time. In HyperFlow, the number of Euler steps is **jointly selected with** the step size during this process, so the hyperparameter budget remains comparable and HyperFlow is not advantaged by extra search. As we show in the sensitivity analysis added in **Figure 5** of the revised paper, HyperFlow has a **stable performance region** and consistently outperforms direct transfer and head-tuning across a broad range of step sizes and step counts.
> >
> > Regarding the “vanishing gradient” part of the question: in our training data we collect **fixed-length trajectories** (50 gradient steps) that are not guaranteed to be fully converged, so the ground-truth gradients used to supervise the drift network have not vanished at the final point. Consequently, we do not enforce the learned drift to go to zero at the last step, and in practice it does not. Empirically, the drift norms tend to decrease along the trajectory but remain non-zero; if we continue integrating beyond the selected number of steps, we sometimes observe a slight loss increase at the tail that the reviewer noted in Figure 4. The validation-based choice of n_steps is precisely meant to stop before this overshoot region, in the range where HyperFlow reliably improves the loss.

---

> > > ### Author Response · Authors · 2025-11-28
> > >
> > > > **Q3**. The hyper-network is trained to predict a scaled one-step update.
> > >
> > > **A4**. Yes, for the linear-flow variant (HyperFlow-L), the reviewer’s interpretation is correct: it effectively learns a scaled one-step update. However, we strictly treat HyperFlow-L as a baseline to demonstrate why simple updates are insufficient. Crucially, our main contributions—**HyperFlow-PL (Piecewise-Linear) and HyperFlow-C (Cubic)**—go beyond this limitation. They model non-linear optimization trajectories where the update direction changes dynamically over time. As shown in our results, these variants significantly outperform the linear baseline, proving that capturing complex, curved update dynamics is essential for high accuracy in few-shot adaptation.
> > >
> > > > **Q4**. HyperFlow method sometimes improves upon bias-tuning.
> > >
> > > **A5**. We hypothesize that this improvement stems from the **implicit regularization provided by amortized inference**. Bias-tuning optimizes a specific task from scratch and can easily overfit to the noise in the few support examples. In contrast, HyperFlow learns a shared update rule constrained to work well on average across thousands of meta-train tasks. This constraint prevents the model from memorizing task-specific idiosyncrasies, effectively "smoothing" the optimization landscape. This leads to a more robust update that generalizes better to the query set—a phenomenon consistent with observations in wider meta-learning literature where amortized methods often outperform per-instance optimization in low-data regimes.
> > >
> > > > **Q5**. Instability of the Euler updates in Figure 4.
> > >
> > > **A6**. The behavior at the tail of Figure 4 reflects the standard trade-off in numerical integration of learned vector fields. Since the drift network is an approximation, small prediction errors can accumulate over long integration horizons. If the solver runs far beyond the optimal point, this cumulative deviation can cause the trajectory to drift, leading to the slight loss increase observed. However, this is functionally equivalent to the "overfitting" regime in standard gradient descent, where training for too many steps degrades test performance. In both cases (fine-tuning and HyperFlow), the solution is identical: we use a validation set to select the optimal stopping point (number of steps). As shown in our results, within the selected operating region, the trajectory is stable and consistently improves performance.
> > >
> > > > **Q6**. Recovery of the optima from the training episodes.
> > >
> > > **A7**. Yes, we have checked this. On meta-train tasks, we start from the same initialization as the recorded fine-tuning trajectories and run Euler integration with the trained drift network. In general, HyperFlow does not exactly recover the final parameter point of the original gradient-descent run: after some number of steps, the trajectory starts to deviate due to approximation error in the learned drift and numerical error accumulation in the ODE solver. Using smaller step sizes and more steps delays this deviation and brings the final point closer to the original optimum, but does not perfectly coincide with it.
> > >
> > > However, HyperFlow still consistently **reduces the loss substantially relative to the initialization** and reaches parameter configurations whose losses are close to those of the fine-tuned endpoints. This is expected, since the drift network is a finite-capacity model trained amortized over many tasks and timesteps; it is optimized to produce a good **approximate optimizer** rather than to reproduce each training trajectory exactly. In fact, forcing exact recovery of every gradient-descent step would likely lead to overfitting the training episodes and hurt generalization to new tasks.
> > >
> > > > **Q7**. Eq.6 should be the time derivative of Eq.5?
> > >
> > > **A8**. Yes, the reviewer is correct: Eq. 6 is intended to denote the time derivative of the path defined in Eq. 5. We thank the reviewer for catching this and have fixed the equation with the proper multiplicative constant (1/T) in the revised version.

---

### Official Review · Reviewer_7DKA · 2025-11-01

**Soundness:** 3
**Presentation:** 3
**Contribution:** 3
**Rating:** 4
**Confidence:** 4

**Summary:**

This paper proposes HyperFlow, a gradient-free alternative to test-time fine-tuning for few-shot classification. The key idea is to emulate gradient descent as an ODE: train a lightweight conditional drift network that predicts task-conditioned parameter velocities from the support set, then adapt by numerical integration without computing gradients of the target model. To keep the problem tractable, HyperFlow only updates a PEFT subset of parameters, learned from offline trajectories simulated on meta-train tasks and interpolated either linearly, piecewise-linearly, or via cubic splines. On Meta-Dataset and CD-FSL, HyperFlow improves OOD performance over direct transfer and sits between direct transfer and full fine-tuning in accuracy.

**Strengths:**

1. Clear core idea: reinterpret gradient descent as an ODE and learn a task-conditioned drift field to perform adaptation with forward passes only.

2. Practical efficiency: compelling compute/memory analysis showing 0.1% FLOPs vs full/bias tuning and ~7–10% of their peak memory, with only ~0.8× overhead vs direct transfer.

3. Solid empirical coverage: results across Meta-Dataset (10 datasets) and CD-FSL (4 datasets) with 600 episodes each; multiple shots; in-domain vs OOD breakdown.

**Weaknesses:**

1. Dependence on offline trajectories: Training requires generating 40k fine-tuning trajectories with multiple random initializations; this offline cost and storage footprint are not quantified and may be non-negligible.

2. Limited backbones and tasks: Experiments focus on ViT-Small ProtoNet for classification; no analysis on larger encoders or other task, such as few-shot classification for CLIP fine-tuning (coop, cocoop etc).

3. Hyperparameter selection: Step counts/step sizes for ODE solving are selected similarly to LR tuning in fine-tuning; more detail on this selection protocol and its sensitivity would help.

**Questions:**

1. Trajectory cost: What are the wall-clock time / GPU hours and storage size for generating the 40k trajectories? Could a smaller set suffice without hurting performance?

2. Backbone generality: Can the conditional drift network trained on ViT-Small generalize to larger ViTs or ConvNeXt without retraining? What fails in cross-backbone transfer?

---

> ### Author Response · Authors · 2025-11-28
>
> > **W1/Q1**. Dependence on offline trajectories / Trajectory cost / Performance with a smaller set
>
> **A1**. We appreciate the reviewer’s request for a detailed breakdown of the offline computational and storage costs. We have added **Table 3** in the revised paper to quantify the wall-clock time and storage footprint strictly. Our analysis confirms that these costs are modest relative to the meta-training stage and can be reduced to negligible levels if needed.
> We first clarify that the primary design goal of HyperFlow is to minimize **test-time adaptation cost** for deployment on resource-constrained devices. The offline trajectory collection is a **one-time server-side procedure**, analogous to the pre-training and meta-training stages in the standard PMF pipeline. In realistic deployment scenarios, this one-time offline investment enables the model to perform gradient-free adaptation for millions of user-specific tasks. Therefore, while we acknowledge the existence of an offline cost, the relevant metric for comparison is the amortized per-episode cost, under which HyperFlow provides a massive efficiency gain over standard fine-tuning.
>
> Regarding the specific costs, we find them to be **modest** when compared to the meta-training stage. As shown in Table 3, the combined process of collecting 40k full trajectories and training the drift network introduces a computational overhead of approximately **25%** relative to the time already required for the backbone's meta-training stage. Similarly, storing the trajectory vectors adds only **~11%** storage overhead compared to the size of the meta-training image dataset. This efficiency is enabled by our strategic choice to target the bias parameter subset, which significantly reduces the dimensionality of the trajectory data compared to full-weight matrices. Also note that **HyperFlow-L** does not require intermediate trajectories and further reduces the storage overhead to **~0.4%**.
>
> Crucially, HyperFlow **does not strictly require 40k trajectories** to be effective. As we already showed in **Table 4** of the Ablation Study section, the method still achieves significant performance gain over Direct Transfer with fewer trajectories. Reducing the dataset to just 400 trajectories lowers the computational overhead to **~2%** and the storage overhead to **~0.2%** relative to the meta-training stage. Despite this massive reduction in offline cost, HyperFlow maintains strong OOD performance (e.g., improving over Direct Transfer by **~5.5%** on Meta-Dataset and CD-FSL, see “40 episodes” row of Table 4), effectively retaining the benefits of the method while eliminating the computational burden.

---

> ### Author Response · Authors · 2025-11-28
>
> > **W2**. Limited backbones and tasks
>
> **A2.** We appreciate the reviewer’s suggestion to explore larger encoders and alternative architectures. HyperFlow is conceptually **architecture-agnostic**: the drift network operates on a selected parameter subset and a task representation from the support set, and the framework does not assume a particular backbone. With an appropriate choice of target parameters, the same procedure can be applied to any encoder to enable gradient-free adaptation at test time.
>
> To empirically support this, we have added experiments with a **ViT-L** backbone, which has roughly **14×** more parameters than the ViT-S backbone. The results are incorporated in **Table 1-2** and **Figure 3** in the revised paper. With this larger backbone, we observe the same qualitative trend as with ViT-S: direct transfer performs worst on cross-domain tasks, gradient-based fine-tuning achieves the best accuracy but with high test-time FLOPs and memory, and HyperFlow consistently sits in between. In particular, HyperFlow **fully benefits from the increased backbone capacity**—it achieves almost the same OOD performance improvement over direct transfer as full/bias fine-tuning does. Moreover, Figure 3 shows that the efficiency advantage of HyperFlow becomes more pronounced with the larger backbone: full-tuning and bias-tuning require about 14× FLOPs and 5–6× peak memory compared to ViT-S, whereas HyperFlow requires only about 10× FLOPs and 4.5× memory. This is because HyperFlow effectively decouples the depth-dependency among target parameters and processes them in parallel.
>
> In **Table 6-7** and **Figure 6** in the appendix, we also report results with a **ConvNeXt-S** backbone, demonstrating that HyperFlow is applicable to convolutional architectures that do not use attention/qkv. We again observe the same “direct transfer < HyperFlow < fine-tuning” pattern, supporting the generality of the HyperFlow framework beyond a single backbone family.
>
> We agree that extending HyperFlow to CLIP-style vision–language backbones and few-shot tasks such as CoOp/CoCoOp is an interesting direction. However, CLIP-based classification typically requires additional design choices (prompt engineering, strong regularization, etc.) that are orthogonal to our current focus. We therefore leave a systematic HyperFlow+CLIP study for future work, and instead provide evidence that the proposed approach generalizes across substantially different vision backbones (ViT-S, ViT-L, and ConvNeXt-S) within the PMF-ProtoNet setting.
>
>
> > **W3**. Hyperparameter selection
>
> **A3**. We appreciate the reviewer’s comment about clarifying the hyperparameter selection protocol. Our goal is to treat HyperFlow and gradient-based fine-tuning symmetrically, and we therefore use exactly the same validation budget and a closely analogous search procedure to PMF’s learning-rate tuning.
>
> Concretely, for each dataset we reserve a small set of validation tasks (5 episodes) prior to evaluation. For gradient-based fine-tuning (bias-tuning and full-tuning), we follow Hu et al. (2023) and run a grid search over learning rates; for each candidate learning rate we train with a fixed number of gradient steps and select the one that maximizes mean validation accuracy, then use that setting for all test episodes. For HyperFlow, we perform the ODE analogue of this procedure. We define a small grid of candidate Euler step sizes. For each step size, we integrate up to 50 steps and, using the same validation episodes, select the number of steps that yields the best mean validation accuracy. Finally, we choose the (step_size, n_steps) pair that performs best on validation and fix it for all test episodes; no per-episode tuning is performed.
>
> To further address the reviewer’s request for sensitivity analysis, we have added sensitivity plots in **Figure 5**. For both fine-tuning and HyperFlow, we report CD-FSL 5-shot accuracy while (1) varying the learning rate (for fine-tuning) or step size (for HyperFlow) with a fixed number of steps, and (2) varying the number of steps with a fixed learning rate / step size. This analysis shows that HyperFlow has a **stable performance region** and consistently outperforms direct transfer and head-tuning across a broad range of step sizes and step counts, supporting that our gains do not rely on brittle hyperparameter choices.

---

> > ### Author Response · Authors · 2025-11-28
> >
> > > **Q2**. Backbone generality
> >
> > **A4**. We train HyperFlow separately for each backbone we deploy it with, exactly as one normally re-trains a meta-learner when changing backbone capacity or architecture. Therefore, we do not claim cross-backbone transfer of a drift network trained on ViT-Small to larger ViTs or ConvNeXt, and we do not rely on such transfer in any of our experiments.
> >
> > The learned mapping from (support set, current parameters, time) to parameter velocities is inherently **backbone-specific**: (1) the dimensionality and structure of the target parameter subspace (e.g., qkv biases vs. ConvNeXt block biases) differ across models, and (2) the gradient flow statistics (layerwise scales, correlations, effective step sizes) change with architecture and depth. Applying a ViT-S drift network directly to a different backbone would therefore misalign both the input parameter space and the output velocity space; this is precisely why we treat HyperFlow as a per-backbone meta-optimizer rather than a universal cross-backbone module.
> >
> > However, as discussed in the response **A2**, HyperFlow does **generalize across backbones when retrained**: we re-run trajectory collection and drift training for ViT-L and ConvNeXt-S, and observe the same qualitative pattern (“direct transfer < HyperFlow < fine-tuning”) with only a modest additional offline cost relative to meta-training each new backbone (≈25% for the 40k-trajectory setting, and ≈0.25% when using only 400 trajectories). In this sense, backbone generalization is achieved via re-meta-training on the new backbone, not via zero-shot cross-backbone reuse of a single drift network. Exploring such universal cross-backbone drifts is an interesting, but out-of-scope, direction for future work.

---

### Author Response · Authors · 2025-11-28
**General Response to the Reviewers**

We thank the reviewers for their constructive feedback and for recognizing several strengths of our work, including the **clarity of the method** (7DKA, 6hQt, Vcrg), the **efficiency of the approach** (7DKA, 6hQt, Vcrg, M6KB), and the **systematic experimental design** (7DKA, 6hQt, Vcrg). We also apologize for the delays in submitting our rebuttal, which was due to completing all experiments requested by the reviewers, and we appreciate the reviewers’ time in considering our responses.

In response to the comments, we have revised the paper with four main updates. First, we added Table 3 to **quantify the offline costs** (compute and storage) and expanded the corresponding discussion in Section 5.3. Second, we validated our method on significantly larger and distinct architectures—**ViT-L and ConvNeXt-S**—and added their results in the main experiment section and Appendix A. Third, we included a new **LoRA-tuning baseline**, which shows that in our few-shot cross-domain setting, bias-tuning offers a particularly favorable accuracy–efficiency trade-off and serves as a strong PEFT baseline for HyperFlow. Finally, we added **hyperparameter sensitivity analyses** in Figure 5 and discussed these results in Section 5.4.

The revision also includes minor updates to the computation cost analysis: we now measure FLOPs and peak memory at 128×128 resolution instead of 224×224, since fine-tuning ViT-L at 224×224 caused out-of-memory errors. The overall efficiency trends, however, remain unchanged.

---

### Meta-Review · Area_Chair_L33e · 2026-01-06

**Summary:**

This paper focuses on the few-shot learning (FSL) problem. Typical FSL approaches involve three stages: pre-training, meta-training, and fine-tuning. The fine-tuning stage is performed at test time and is often time-consuming due to parameter updates. The authors propose an optimization-free approach that decomposes the fine-tuning stage into two separate phases, with the first phase learning the optimization dynamics offline. In essence, the method shifts the computational burden from test time to offline training. The paper received four reviews with mixed ratings (one positive and three negative).

The AC carefully reviewed the paper, the reviews, and the author responses. The AC believes this is a well-written paper with clear motivation and a systematic design. However, the paper lacks comprehensive experimental validation, a concern also raised by the reviewers. Although the authors provide additional validation across different backbones, the proposed method is claimed to be not only model-agnostic but also method-agnostic, serving as a plug-in module for any approach involving pre-training, meta-training, and fine-tuning. Despite this claim, the experimental evaluation is limited to ProtoNet. Furthermore, the architecture of HyperFlow (Figure 2), which serves as a critical component of the method, is not sufficiently ablated. The scope of the main comparisons is also limited. In addition, the proposed approach leverages gradient-free benefits that are primarily associated with a specific line of FSL research. To better highlight the contribution, the authors should include comparisons with other lines of work, such as metric-based methods and hypernetwork-based approaches.

Overall, this is a solid paper with strengths recognized by the reviewers. However, due to the limited experimental validation of its broader applicability, the AC recommends rejection. The authors are encouraged to incorporate the reviewers’ feedback and consider resubmission in the future.

**Reviewer Concerns:**

The major concerns can be summarized as follows: (i) cost analysis and the impact on the generated offline trajectories [7DKA, Vcrg, M6KB]; (ii) limited experimental validation in terms of base methods, backbones, and tasks [7DKA, Vcrg]; (iii) limited experiments on the proposed hypernetwork, including generalization and varying numbers of ODE steps [6hQt, Vcrg, M6KB]; (iv) lack of theoretical guarantees [Vcrg]; and (v) an accuracy gap compared to the baseline [M6KB, 6hQt].

Based on the author responses, the AC believes that concerns (i) and (iv) have been adequately addressed. However, concerns (ii), (iii), and (v) remain unresolved.

**Reviewer Scores:**

Reviewers [7DKA] and [6hQt] may increase their ratings, but are unlikely to assign a score of 6. The AC believes that they may still lean toward rejection. Reviewers [Vcrg, M6KB] are expected to maintain their current ratings.

---

### Decision · Program_Chairs · 2026-01-26

Reject